# Spatio-temporal dynamics enhance cellular diversity, neuronal function and further maturation of human cerebral organoids

Pelin Saglam-Metiner[1], Utku Devamoglu [1], Yagmur Filiz [1], Soheil Akbari[2], Goze Beceren[1], Bakiye Goker[3], Burcu Yaldiz[1], Sena Yanasik [1], Cigir Biray Avci[3,4], Esra Erdal[2,5] & Ozlem Yesil-Celiktas [1✉]

The bioengineerined and whole matured human brain organoids stand as highly valuable three-dimensional in vitro brain-mimetic models to recapitulate in vivo brain development, neurodevelopmental and neurodegenerative diseases. Various instructive signals affecting multiple biological processes including morphogenesis, developmental stages, cell fate transitions, cell migration, stem cell function and immune responses have been employed for generation of physiologically functional cerebral organoids. However, the current approaches for maturation require improvement for highly harvestable and functional cerebral organoids with reduced batch-to-batch variabilities. Here, we demonstrate two different engineering approaches, the rotating cell culture system (RCCS) microgravity bioreactor and a newly designed microfluidic platform (μ-platform) to improve harvestability, reproducibility and the survival of high-quality cerebral organoids and compare with those of traditional spinner and shaker systems. RCCS and μ-platform organoids have reached ideal sizes, approximately 95% harvestability, prolonged culture time with Ki-67 + /CD31 + /β-catenin+ proliferative, adhesive and endothelial-like cells and exhibited enriched cellular diversity (abundant neural/ glial/ endothelial cell population), structural brain morphogenesis, further functional neuronal identities (glutamate secreting glutamatergic, GABAergic and hippocampal neurons) and synaptogenesis (presynaptic-postsynaptic interaction) during whole human brain development. Both organoids expressed CD11b + /IBA1 + microglia and MBP + /OLIG2 + oligodendrocytes at high levels as of day 60. RCCS and μ-platform organoids showing high levels of physiological fidelity a high level of physiological fidelity can serve as functional preclinical models to test new therapeutic regimens for neurological diseases and benefit from multiplexing.

[1] Department of Bioengineering, Faculty of Engineering, Ege University, 35100 Izmir, Turkey. [2] Izmir Biomedicine and Genome Center (IBG), Dokuz Eylul University Health Campus, 35340 Izmir, Turkey. [3] Department of Medical Biology, Faculty of Medicine, Ege University, Bornova, Izmir 35100, Turkey. [4] Department of Stem Cell, Institute of Health Science, Ege University, Bornova, Izmir 35100, Turkey. [5] Department of Medical Biology and Genetics, Faculty of Medicine, Dokuz Eylul University, 35340 Izmir, Turkey. ✉email: ozlem.yesil.celiktas@ege.edu.tr

Understanding of brain generation, maturation, and functional development is vital for comprehending evolutionary brain development, as well as developing treatment strategies for neurodegenerative and neurodevelopmental diseases[1,2]. Although animal models are useful for mechanical examination, recapitulation of human brain development is significantly limited due to differences at cellular and molecular levels[3]. Moreover, much of our current knowledge of the human brain is based on the analysis of postmortem or pathological specimens inappropriate for experimental manipulation[4]. Lately, induced pluripotent stem cells (iPSC), having the capacity to regenerate continuously with reprogramming factors (OCT4, SOX2, c-MYC, and KLF4) and differentiate into all cell types in the body, have opened a new avenue for studying human neural development[5]. With the first iPSC-derived 3D cerebral organoid generation protocol published by Lancaster et al.[6], cerebral organoids have become groundbreaking with the ability to self-organize, form more complex structures, transform into different progenitor regions, mimic the cell type composition/tissue organization of the embryonic brain and resemble the spatial structure of the brain[7]. Besides, similar to the process of brain formation and development in the embryonic period, cerebral organoids can closely mimic the human brain by generating epigenomic and transcriptional programs such as signals associated with cell diversity, clarification of functional synapses, formation of dendritic spines and establishment of self-activated neuronal networks[8,9].

Conventional cerebral organoid maturation approaches using spinners[10] and shakers[11], promote self-neuronal organization and formation of cerebral ventricular spaces by providing agitation. However, these methods have major limitations, including altered nutrient diffusion and apoptotic zones due to lack of vascularization, immune cell deficiency, matrigel dependency, low reproducibility, scalability, and high variability of the induced brain components and cells[12]. Different strategies have been developed to prevent these limitations such as by co-culturing with human umbilical vascular endothelial cells (HUVECs)[13] and human iPSC-derived endothelial cells not only to enhance oxygen/nutrient diffusion but also to leverage neural differentiation, migration, and circuit formation during development[14]. Other strategies focus on recapitulating the cellular microphysiological environment with decellularized brain extracellular matrix (ECM)[15] or synthetic ECM-like matrix to promote neural and glial differentiation during brain organogenesis[16,17]. Although not fully resolved, low reproducibility, scalability and high variability problems of brain organoids have been tried to overcome with different organoid-on-chip platforms[18,19] or novel bioreactor systems such as single-use vertical-wheel bioreactors[20]. Despite the progression of organoid technologies, hemodynamic forces such as shear stress, cyclic stretching and fluid distribution regulating mechanosensitive pathways in cerebral organoids are still not clearly understood which is another critical limitation. However, mechanosensitive signals associated with the cerebrospinal fluid flow during embryogenesis are known to regulate important structural adaptations such as the orientation of the embryonic shape with the effect of left-right asymmetry, the polarization of radial glial cells, differentiation/functionalization of neural stem cells and later tissue folding events[21,22]. Thus, there is a need to develop platforms with the use of biology and engineering principles that can emulate the spatiotemporal dynamics and cell–cell/cell–environment interactions. Herein, we analyze the effects of various flow characteristics both computationally and experimentally on the maturation of cerebral organoids and elicit the best strategy for engineering iPSC-derived 3D cerebral organoids at molecular, cellular, and functional levels. The microfluidic platform (µ-platform) allows gravity-driven laminar flow that mimics fluid flow existing in the cerebrospinal and interstitial spaces and enhances cell–cell crosstalk, oxygen supply, and nutrient/waste exchange[7,23], leading to longer-lasting organoids. Furthermore, unusually provided microgravity and laminar flow conditions by the horizontally rotating cell culture system (RCCS) bioreactors exhibit homogeneous hydrodynamic forces facilitating long-term precise control of cell–cell interactions and allow high harvestability and reproducibility of matured cerebral organoids. RCCS, inspired by extraordinary conditions in space technologies[24] can be a unique tool in organoid research due to reduced shear stress, increased mass transfer, and decreased cell damage compared to traditional systems[25]. We hypothesize that microgravity and gravity-driven laminar flows enhance the maturation of cerebral organoids leading to hierarchically complex spatial networks recapitulating features of human embryonic cortical development.

## Results

### RCCS and µ-platform dynamic systems exhibit less shear stress during organoid maturation compared to spinner and shaker systems.

The hemodynamic effects on organoids were simulated using hydrodynamic forces, which occurred by shaking, spinning, and rotating in microgravity and laminar flow, and analyzed by flow simulations using COMSOL Multiphysics. Considering the fluid flow profile differences between the approaches, a 3D computational model was developed using turbulent flow principles with the $k–\varepsilon$ equation for shaker and spinner, laminar flow principles with Navier–Stokes equation for µ-platform, and RCCS. Initial conditions of rotational speed and fluid flow rates in the simulations were determined to allow maximum nutrient and oxygen transfer as 100–120 RPM in the shaker, 50-60 RPM in the spinner[6], 100–800 µL/min in the µ-platform[26] and 10–15 RPM in RCCS. Large differences were observed in the shaker between the maximum and minimum shear stress values of 1.57 and $1.93 \times 10^{-2}$ Pa due to the maximum and minimum velocities of 0.42 and $7.24 \times 10^{-2}$ m/s. Similar flow profiles were observed in the spinner, where maximum and minimum shear stress values were 0.96 and $9.97 \times 10^{-4}$ Pa corresponding to the velocities of 0.10 and $8.65 \times 10^{-5}$ m/s. Although maximum shear stress values for the spinner were in the same order of magnitude, the computed value for the shaker was one order of magnitude higher compared to the reported data[27], which might be due to the finite element meshes and flow conditions used in the computational fluid dynamics (CFD) module. As for the laminar flow profiles, the maximum and minimum shear stresses on the µ-platform were observed as $7.72 \times 10^{-}$ and $5.13 \times 10^{-7}$ Pa. While this partially high shear stress difference was observed at the connection areas of the organoid chambers, the average shear stress effect on the organoids and in the chambers is approximately $4 \times 10^{-4}$ Pa. Although higher minimum and maximum velocities were noted in RCCS compared to µ-platform as $1.05 \times 10^{-2}$ and $3.14 \times 10^{-2}$ m/s, the minimum, and maximum shear stress values were lower as $2.69 \times 10^{-5}$ and $2.32 \times 10^{-4}$ Pa (Fig. 1a). As expected, the maximum shear stress distribution was observed at the boundaries of the shaker and around the impellers of the spinner, leading to non-homogeneous distributions (Fig. 1b, c). However, more homogeneous shear stress distributions were noted in the µ-platform and RCCS owing to the laminar flow characteristics (Fig. 1d, e). The initial values of computed rotational speeds and fluid flow rates were used for maturation experiments with slight increases to avoid coalescence and aggregation of organoids growing in size, yet still remaining in the computed ranges.

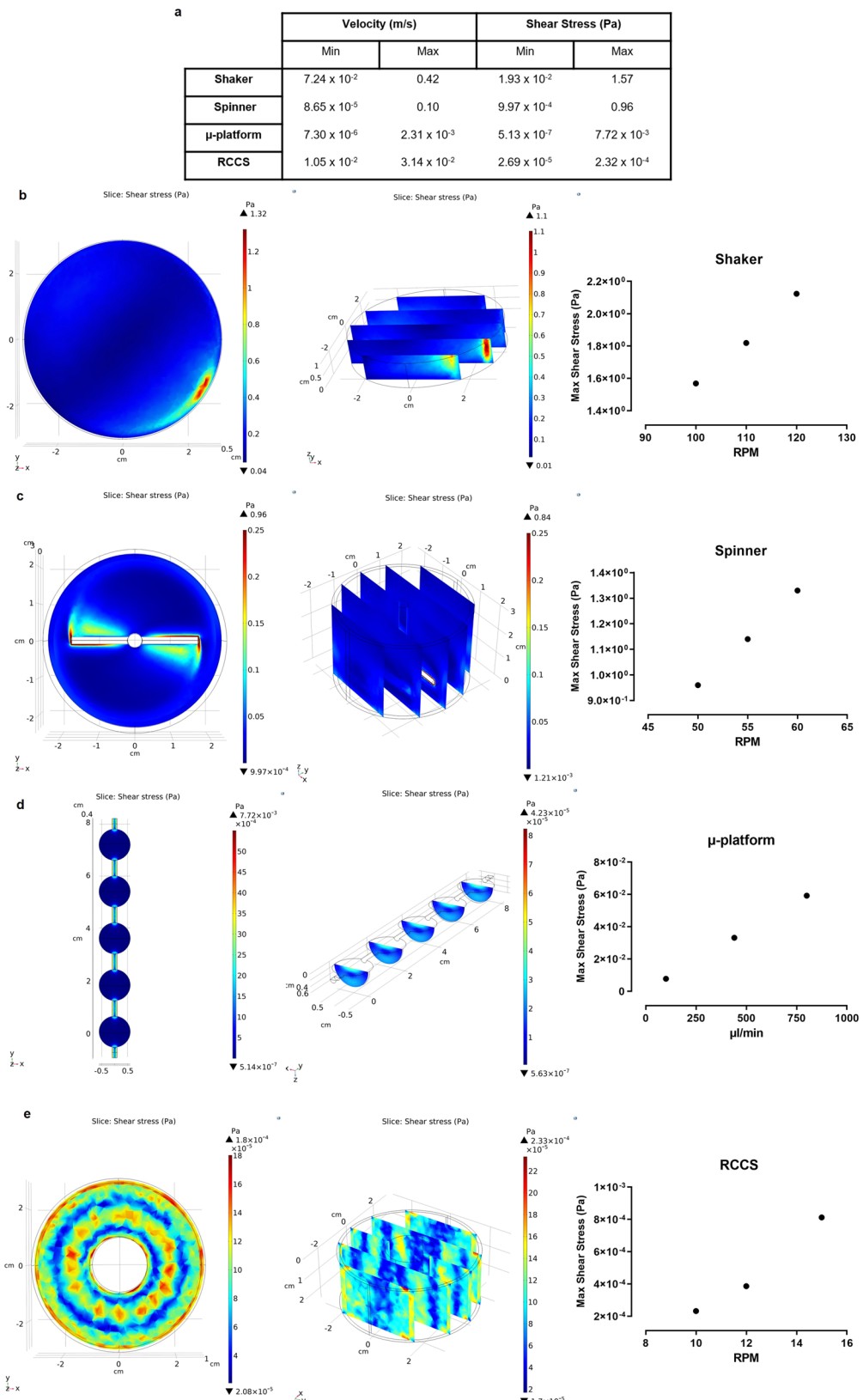

**Fig. 1 Results of computational fluid dynamics (CFD) simulations using finite element commercial code COMSOL Multiphysics. a** Min and max values of velocity and shear stress at initial flow conditions. Numerical simulations of shear stress for; **b** shaker, **c** spinner, **d** μ-platform, and **e** RCCS-microgravity bioreactor.

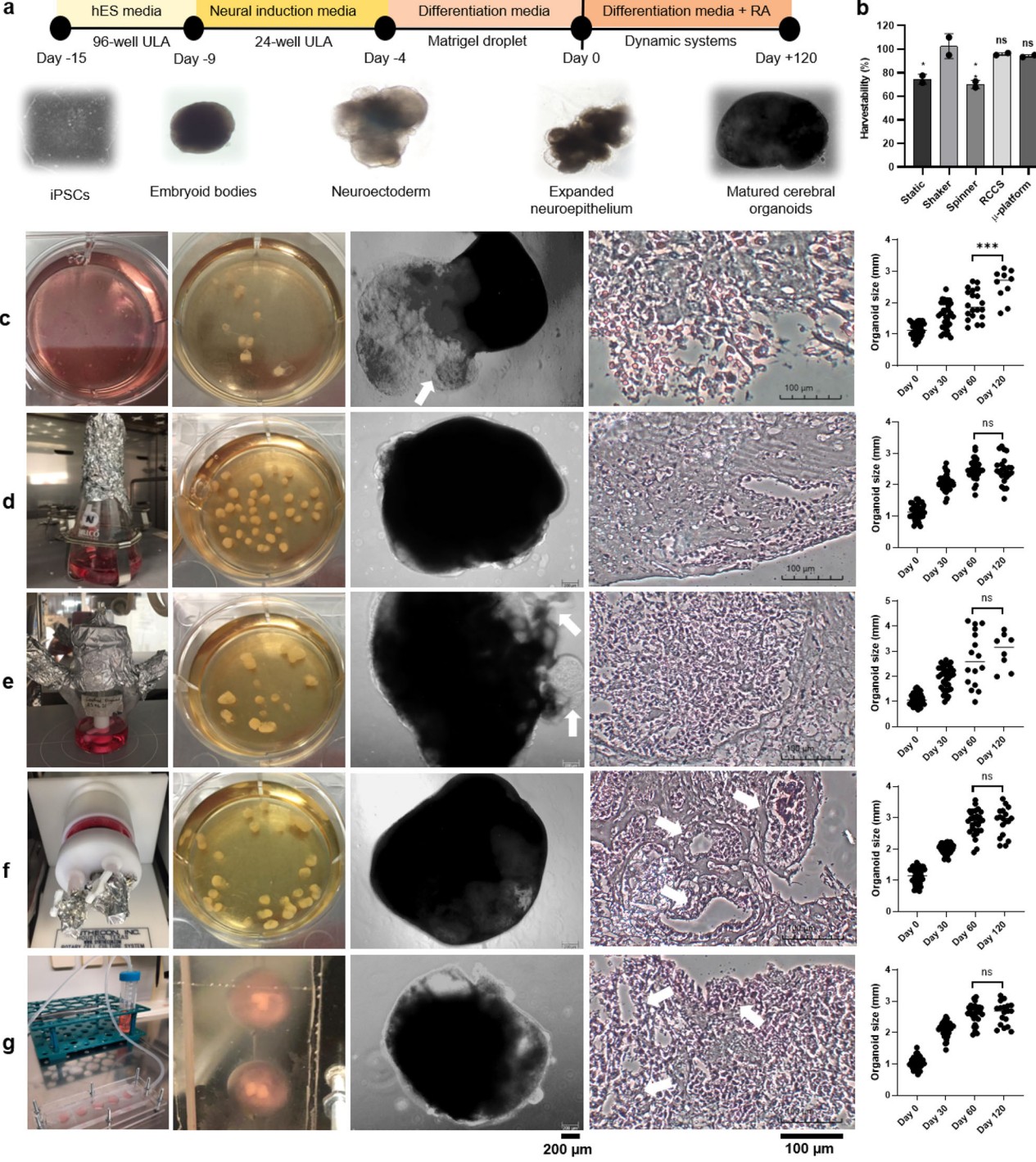

**Fig. 2 iPSCs derived cerebral organoid formation process. a** Generation steps. **b** Percentage of organoid harvestability results (shaker versus other systems, *$p = 0.0103$ for static, **$p = 0.0054$ for a spinner, ns; $p = 0.5873$ for RCCS and ns; $p = 0.4343$ for μ-platform, $n = 2$ for independent replicates = 2). Maturation steps by **c** static and dynamic systems as **d** shaker, **e** spinner, **f** RCCS microgravity bioreactor, and **g** μ-platform. From left to right, the system overviews, macroscopic images, bright-field microscopic images (white arrows indicate off-target cell differentiation in preserved matrigel matrix) (Zeiss Axio Vert.A1) of organoids and H&E stained (Motic, AE31E) organoids matured in different systems at day 60 (white arrows indicate sequentially organized cell nucleus and neural rosette-like structures) (scale bars = 200 μm for bright-field and 100 μm for H&E staining images, independent replicates = 3), and plots of organoid sizes (day 120 versus day 60, ***$p = 0.0002$ for static, ns; $p = 0.9582$ for shaker, ns; $p = 0.5787$ for a spinner, ns; $p = 0.9945$ for RCCS and ns; $p = 0.9918$ for μ-platform, independent replicates = 2) where diameters were measured with the ImageJ program at days 0, 30, 60, and 120 ($n = 40, 33, 20, 10$ for static, $n = 40, 40, 34, 24$ for shaker, $n = 40, 36, 16, 8$ for a spinner, $n = 40, 40, 30, 18$ for RCCS, and $n = 40, 38, 28, 18$ for μ-platform, all for on day 0, 30, 60, 120, respectively).

**Laminar flow increases the reliable production of high-quality organoids with the highest harvestability, reproducibility, and reduced batch-to-batch variabilities**. During the cerebral organoid generation process (Fig. 2a), the hESC medium was changed by half every other day for iPSCs cultured in 96-well ultra-low-attachment (ULA) plates in the first step (on the day −15). As the embryoid body (EB) mimicked the germ layer, microscopical observation was carried out for about 5–6 days until bright and smooth structures with sharp borders greater than 350–400 μm in diameter were obtained (on the day −9). Subsequently, EBs transferred to 24-well ULA plates were placed in a neural induction medium and monitored for another 4–5 days with medium changes every other day until radially organized, optically translucent neuroectodermal structures were formed, which were brighter on the outside, indicating neuroectodermal differentiation. At day 11 of differentiation, the tissues were embedded in matrigel drops and cultured in 6-well suspended plates in cerebral organoid differentiation medium without vitamin A for an additional 3–4 days, until the formation of more enlarged neuroepithelial buds containing fluid-filled spaces (on the day −4). On the 15th day of culture, generated organoids (derived from two different passage numbers of iPSCs at two different times) were transferred to RCCS, μ-platform, spinner, and shaker, along with static culture as the control group (considered as day 0 of maturation), in order to determine the best dynamic system for physical and functional maturation of cerebral organoids, which were cultured for 120 days. Organoid size distribution variability is known to be a limitation of traditional human brain organoid protocols, as well as cellular composition and organization. Therefore, organoid reproducibility and batch-to-batch variability were first examined in terms of harvestability, macroscopic–microscopic observations, and organoid size distribution (Fig. 2b–g, Supplementary Data 1a, b). As such, cellular composition and cellular organization were assessed with cell sorting, immunostaining, western blot, qRT-PCR, TUNEL, and glutamate secretion analyses for organoids sampled on various days (30, 60, and 120th days). When the macroscopic-microscopic images taken on certain days of the cultures were examined, non-homogeneous shapes and fluid-filled cystic structures (transparent regions resemble cystic formation associated with off-target cell differentiation in preserved matrigel matrix, indicated by white arrows) without well-developed neuroepithelium[6] were observed in static culture. As for organoids matured in laminar flow directed μ-platform and RCCS, both homogeneous sizes, self-organization, and ideal cerebral ventricular structures were well distinguished, compared to other dynamic systems. Although the largest organoid sizes could be reached in the spinner, because of fragmentation/aggregation caused by the effect of complex mechanical agitation and undirected eddies, organoid structures with non-homogeneous sizes-shapes and cystic structures (indicated by white arrows) were observed. Besides, hematoxylin and eosin (H&E) staining showed intact cell nuclei within relatively small cells homogeneously well-arranged throughout the organoids, except for static culture. In terms of tissue structure, while the most dispersed organoid structure was observed in static culture and the most uniform cell distribution (unorganized) was observed in the shaker, the well-defined multi-patterned organoid morphologies with self-organization and desired cellular architectures in the inner–outer regions[6,12,28] were observed more in the RCCS microgravity bioreactor (sequentially organized cell nucleus and neural rosette-like structure indicated by white arrows), followed by μ-platform and spinner systems (Fig. 2c–g).

Herein, the microscopic images taken at desired days of organoid maturation in dynamic systems were analyzed by the ImageJ program to create organoid size graphs. Each system was started simultaneously with generally 40 organoids derived from the same passage of iPSCs (with 2 independent replicates of different iPSCs passages at 2 different times). Only the μ-platform system was operated as 4 separate parallel cultures with 10 organoids/platforms. For industrial scalability and standardization, the percentage of harvestability with average size on the last day of the cultures was calculated considering disintegration–deterioration–dispersion–aggregation of cultured organoids. In the static culture, the harvestability was measured as 74.5% ± 4.3 where the organoids could be harvested at an average size of 2.56 ± 0.49 mm, reaching maximum sizes of 3.1 mm at day 120 with non-homogeneous courses (Fig. 2b, c). Interestingly, in the shaker system, the harvestability was calculated above 100% (102.5% ± 10.6), possibly due to the disintegration and fragmentation of organoids under mechanical agitation. Homogeneous and distinctly sized organoids were harvested at an average size of 2.45 ± 0.42 mm, reaching maximum sizes of 3.16 mm at day 120 (Fig. 2b, d). In a study, cerebral organoids matured in a shaker were reported to be around 350–400 μm at the beginning, reaching 3–5 mm after 2 months and maintaining sizes for 5–6 months[29]. In the spinner, the lowest harvestability was 70% ± 3.5 ($p = 0.0054$, compared to the shaker) on the last day of the culture, organoid sizes were observed in a non-homogeneous course with a wide size distribution. The highest organoid sizes reached 4.21 and 3.87 mm at days 60 and 120, respectively, with average sizes of 3 ± 0.71 mm at day 120 (Fig. 2b, e). Similar results were noted, where cerebral organoids reached ≤3 mm in size at 100 days to produce forebrain, midbrain, and hypothalamus organoids in the spinner[30]. The harvestability of RCCS organoids was 95.8% ± 1.2 (ns; $p = 0.5873$, non-significant compared to the shaker). Homogeneous and comparatively larger-sized organoids with a narrow size distribution were obtained at an average size of 2.87 ± 0.45 mm, reaching maximum sizes of 3.60 mm at day 120 (Fig. 2b, f). On the contrary, hESC-derived forebrain-specific neural-cortical organoids were reported to reach sizes of about 1.4 mm in RCCS-HARVs-SC 10 mL vessel culture at day 49[31]. As for the μ-platform organoids, the harvestability was determined as 94.2% ± 1.2 ($p = 0.4343$, non-significant compared to the shaker) with a narrow size distribution and an average size of 2.61 ± 0.36 mm, reaching maximum sizes of 3.18 mm at day 120 (Fig. 2b, g). Overall, the organoid sizes remained almost constant as of day 60 within all established dynamic systems (statistically non-significant $p > 0.5$, mostly for RCCS $p = 0.9945$). In spite of the low-velocity fields exercised in the RSSC and the μ-platform, nutrient mixing has been efficient to support organoid growth and low shear environment yielded homogeneous size distributions, which is of prime importance for multiplex testing in the pharmaceutical industry.

**Laminar flow enhances neurogenesis/gliogenesis and cellular diversity of the brain organoid.** Cerebral organoids recapitulate many features of the developing cortex, including neurogenesis/gliogenesis and formations of distinct human brain-specific cell types that resemble in vivo human brain development, which is highlighted by changes in levels of various markers associated with differentiation and maturation of neural stem cells/progenitors into different brain cell types (Fig. 3a). To examine the generation of multiple cell types in cerebral organoids matured in different dynamic systems, immunofluorescence staining (IF), western blot (WB), qRT-PCR and fluorescence-activated cell sorting (FACS) analyses were performed and supported with published datasets of RNA expression levels (Human Protein Atlas V21.0 proteinatlas.org) of specific markers in different neuronal cell types[32]. Specific neuronal cell markers were studied at different time points of differentiation at protein and RNA

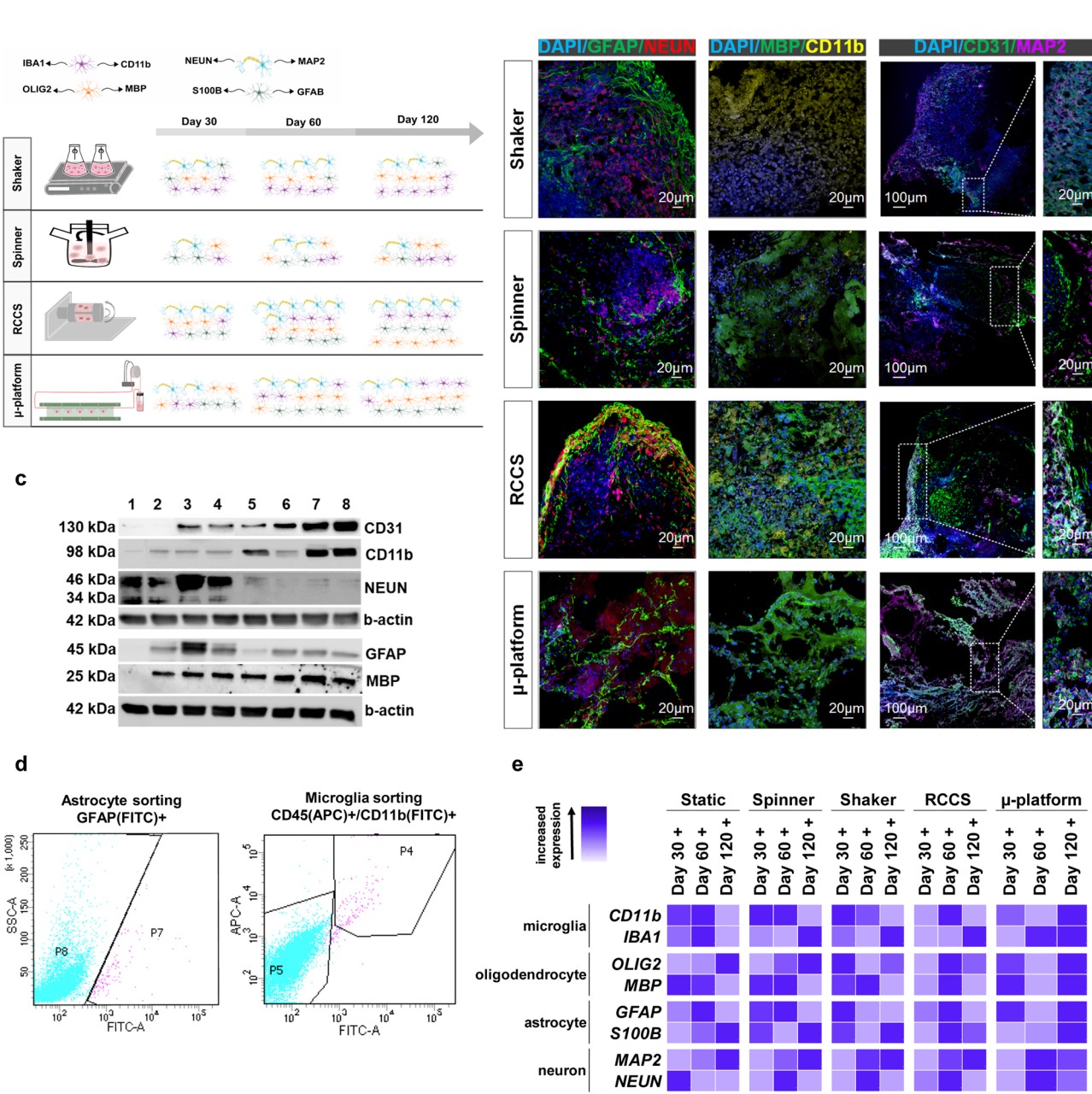

**Fig. 3 Cellular diversity of cerebral organoids. a** A schematic illustration of cell diversity of cerebral organoids matured in dynamic systems. **b** Immunofluorescence staining of specific neuronal/glial/endothelial cell markers (NEUN/GFAP, MBP/CD11b, CD31/MAP2) at 120 days of organoids matured in 4 dynamic systems (scale bars = 100 μm for 10× and 20 μm for 25× magnification images, independent replicates = 3, Zeiss LSM 880). **c** WB analysis for neuronal proteins of cerebral organoids matured in a shaker (1,5), spinner (2,6), RCCS (3,7), and μ-platform (4,8) at day 60 and 120, respectively (independent replicates = 2). **d** FACS analysis of microglia and astrocytes in RCCS organoids on day 60 (independent replicates = 2). **e** Heatmap data of the qRT-PCR analysis of cerebral organoids matured in static, spinner, shaker, RCCS, and μ-platform on day 30, day 60, and day 120 in comparison to generated organoids on day 15 (independent replicates = 2).

expression levels. Glial fibrillary acidic protein (GFAP) positive cells indicating the presence of astrocytes differentiated from radial glial cells[33] have become detectable at day 60 of differentiation (Supplementary Fig. 1) and been markedly positive at day 120 (Fig. 3b, c, Supplementary Fig. 2a), where cells were located specifically at the periphery of the organoids. Astrocyte maturation was observed at enhanced levels in all dynamic systems, with comparatively higher expression levels in RCCS both at day 60 and 120 (Supplementary Fig. 1b, Fig. 3b, c, Supplementary Fig. 2a), with 1.3% ± 0.4 GFAP + cell sorting rate

(Fig. 3d). In addition to GFAP, the astrocyte associated marker S100 calcium-binding protein B (S100B) was examined (Fig. 3e, Supplementary Table 1). Astrocytes are found in the mid to late stages (12–40 gestational week (gw)) in the neurodevelopment process, where the expression of astrocyte markers showed similarities to neurodevelopmental stages[34], suggesting that RCCS promotes astrocyte differentiation at an earlier stage than the μ-platform. Substantial protein expression of neuronal nuclei (NEUN), a marker of mature neurons[35], was observed in all systems as of day 60 of maturation (Supplementary Fig. 1, Fig. 3b, c, e,

Supplementary Fig. 2a, Supplementary Table 1). Interestingly, the expression levels of NEUN started to decrease after day 60, where the lowest expression was at the spinner, while partially high NEUN expression was observed in RCCS on day 120. These results are in line with the neurodevelopmental process in which neurons begin to be generated at 8–10 gw, increased until mid-gestation and then decreased[36,37]. Also, especially in the RCCS, cells expressing GFAP co-localized with those expressing NEUN on day 120, suggesting promotion of neural population generation during maturation. Then, we further assessed the spectrum of cellular labeling with co-staining of mature microglia cell marker, cluster of differentiation 11b (CD11b), a resident brain immune cell associated with neuroinflammation, which plays an important role in the course of many neurodegenerative and neurodevelopmental diseases[38] and myelin basic protein (MBP), expressed on the cytoplasmic surface of mature oligodendrocytes which differentiated from oligodendrocyte progenitor cells[39] (Supplementary Fig. 1a). IF analysis revealed generally elevated expression levels of CD11b and MBP throughout the whole organoid structure in RCCS and μ-platform, whereas dramatically lower levels in spinner and shaker at day 120 (Fig. 3b, c, Supplementary Fig. 2a). Microglia appear in the early stages of brain development, such as 4.5 gw, and continue to proliferate until 14–17 gw[40,41]. However, in our study, while the microglial marker CD11b was observed less on day 60, which corresponds to approximately 8 gw, it showed an increased expression at day 120, approximately in the mid-gestation, just like in vivo human brain development. To further interrogate the presence of microglial cells, the ratio of CD45/CD11b double-positive cells in RCCS organoids was found to be 1.3% ± 0.4 with FACS analysis, indicating the positive effect of microgravity on mesodermal–microglial cell differentiation (Fig. 3d). In agreement with our findings, Ormel et al. reported that CD11b + microglia constituted 5% ± 2.8 of the entire single-cell suspension with MACS analysis, as well as CD11b+/CD45+ microglia population was 0.83% ± 0.3 with FACS for 38–52 days old cerebral organoid[42]. Shiraki et al. showed that the ratio of CD11b/CD45 double-positive PAX6-positive microglia cells was 0.33% ± 0.12 in 4 weeks of differentiation of hiPSC-derived ocular organoids[43]. Additionally, the well-known specific microglial marker, ionized calcium-binding adapter molecule 1 (IBA1), and oligodendrocyte transcription factor 2 (OLIG2)[44] were also examined (Supplementary Fig. 1a). At day 120, while IBA1 up-regulation was noticeable for RCCS and spinner, both CD11b and IBA1 expression levels were upregulated and showed increasing tendency from day 30 to 120 for organoids matured in μ-platform mainly. Besides, OLIG2 and MBP expression levels were upregulated for RCCS and μ-platform organoids on day 120 (Fig. 3e, Supplementary Table 1). Oligodendrocytes appear in the mid-gestation (22 gw) and their functionality increases at 28 gw[45]. In our study, on day 60 corresponding to early neurodevelopment, oligodendrocyte markers were mostly observed in RCCS, and an increase in oligodendrocyte markers was observed in all systems as the culture period extended to 120 days, which corresponds to mid-gestation. These results suggested that the expressions of genes encoding microglial and oligodendrocyte cell surface proteins have been enhanced by laminar flow in general. Thus, RCCS and μ-platform can be preferred over other systems, where glial cells are to be studied in-depth. We next examined the expression of various markers that delineate different cell populations in cerebral organoids and looked into other markers such as microtubule-associated protein 2 (MAP2) and a cluster of differentiation 31 (CD31), related to neurons matured from neural precursor cells[46] and blood vessel-related endothelial cells that provide vascular integrity[47,48], respectively. MAP2 began to be expressed by day 60 with a progressive increase till day 120, generally for all dynamic systems

(Supplementary Fig. 1b, Fig. 3b, c, e, Supplementary Fig. 2a, Supplementary Table 1), suggesting stimulation of neuronal differentiation since it has been previously described that neurons increase from early to mid-gestation. Apart from that, although CD31+ vessel-like structures were shown to be sporadically throughout the tissue in 2-month-old cerebral organoids[48], observation of CD31 expression without any additional growth factors for endothelial cells is highly valuable, since one of the major limitations of cerebral organoids is generally the lack of functional vascular structure[14]. At day 120, the protein expression level of CD31 was low in the shaker and spinner systems, whereas notable upregulations were observed in the whole body of organoids in the RCCS and μ-platform, mainly (Fig. 3b, c, Supplementary Fig. 2a). At early gestation, 8 gw, which refers to day 60, vascularization of the brain starts. At the end of the 8 gw, vascularization progresses and continues until 5 gw. The vascular structure mostly completes its formation in the brain, which reaches functional maturation at 15 gw[49]. High levels of CD31 expression observed on day 120 reveal similarities with mid-gestation in the neurodevelopment process. Taken together, we concluded that prolonged culture time, the presence of hemodynamic forces such as certain levels of shear stress approximately at a magnitude of $10^{-4}$ Pa and fluid distribution regulating mechanosensitive pathways promote neurogenesis and gliogenesis where various brain-associated cell types derived from different layers of embryonic developmental processes such as ectoderm and mesoderm are formed and matured.

**Laminar flow induces complex multi-layered organization, early/late stage brain development, corticogenesis, and various brain regions in brain organoids.** During human embryonic brain development, neurons and glial cells are derived from neural stem/progenitor cells that generate specific brain regions and cortical layers. The derivation is achieved by an asymmetric spatial organization of cells and different components, explained as cell polarity. In the polarization process, the ventricular zone (VZ) formed by neuroepithelial cells is followed by the subventricular zone (SVZ), where apical progenitors form basal progenitors, and then cortical plate (CP) by apical and basal progenitors form neurons[50]. The degree of efficient differentiation is determined by the expression of neurons at different polarized states and progenitor-specific markers. Herein, RCCS organoids exhibited well-defined CP, SVZ, and VZ architecture[51–53] on day 120, where beta-tubulin III (TUJ1), encoding neuron-specific tubulin was expressed in proliferating neuronal cells in the cortex layer and mainly located at the basal side of the ventricle-like structures[54,55], whereas TUJ1+ cells were present throughout the whole constructs mainly for organoids in all dynamic systems at day 60 and 120 (Fig. 4a, b, Supplementary Fig. 2, Supplementary Table 1). Although TUJ1 expression has not differed with respect to protein levels among dynamic systems (Fig. 4c, Supplementary Fig. 2b), statistically much higher RNA upregulations were noted in RCCS and μ-platform on days 30 and 60 (Fig. 4d, Supplementary Table 1). (T-box brain transcription factor 1) TBR1+ cells that differentiate from intermediate progenitor cells[54,56] were more abundant in the spinner and μ-platform organoids at days 30 and 60 with a layer adjacent to TUJ1+ zone (white arrows indicate neural rosette-like structures), and in RCCS organoids mainly at day 120 (Fig. 4, Supplementary Fig. 2b, Supplementary Fig. 3, Supplementary Table 1) in the deep layer of cortical-like plates[51]. The expression level of nestin is directly proportional to neural progenitor cells but varies in differentiated tissues[57]. Indeed, nestin expression was upregulated at both protein and RNA levels in the shaker and RCCS organoids at days 30 and 60 mostly (Fig. 4c, d,

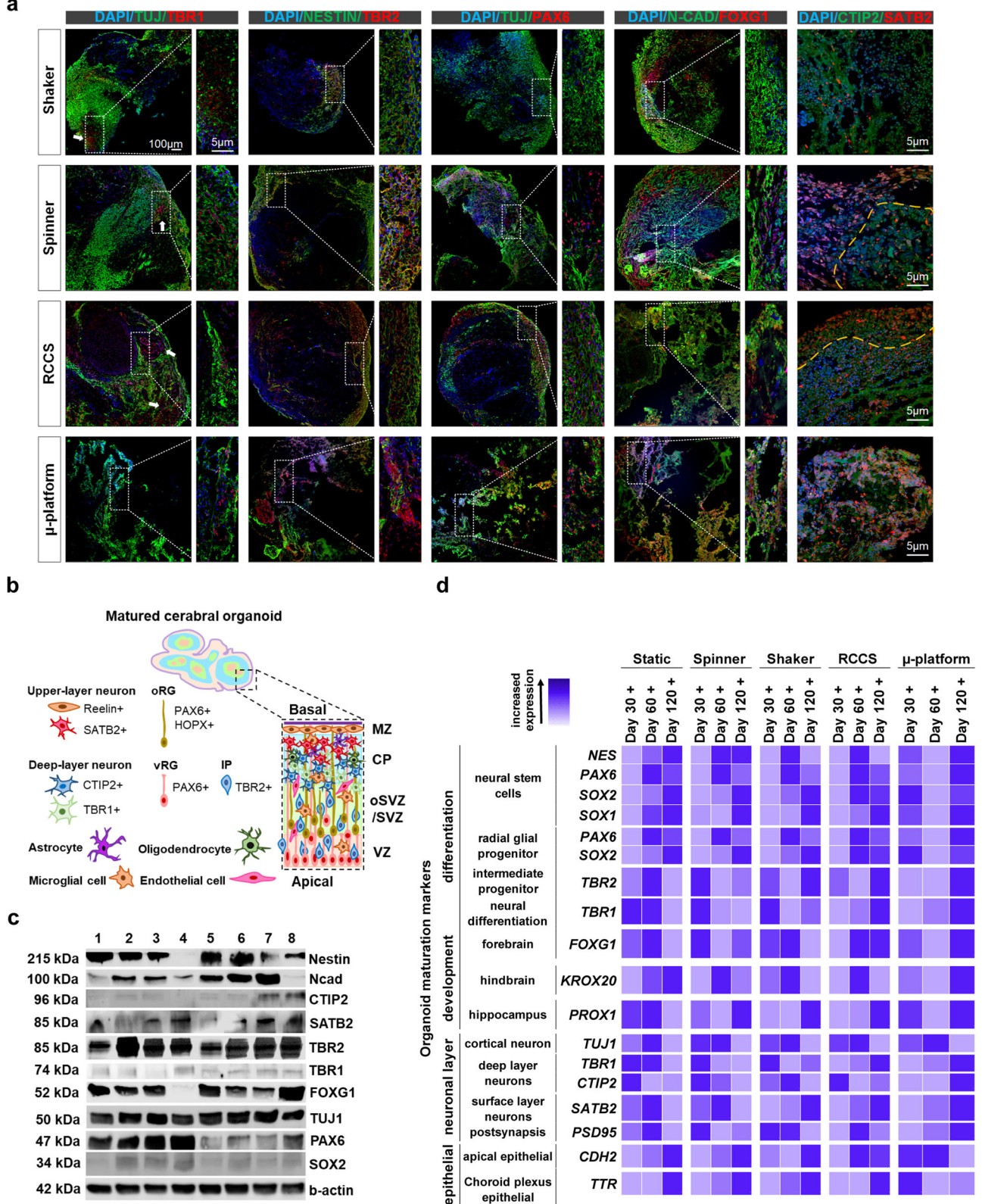

Supplementary Fig. 2b, Supplementary Fig. 3, Supplementary Table 1), and all dynamic systems at day 120 (Fig. 4). Moreover, the co-localization of nestin and intermediate progenitor cell marker, (T-box brain protein 2) TBR2 indicate that the cerebral organoids possessed well-defined progenitor zone organization and neural identity[58] for all dynamic systems at day 120 but mostly in μ-platform and RCCS (Fig. 4a–c, Supplementary

Fig. 2b, Supplementary Fig. 3, Supplementary Table 1). Paired box protein 6 (PAX6) is a transcription factor expressed in neural stem cells along with ventral forebrain radial glial progenitor cells and has an important role in the differentiation of radial glial cells located in different regions of the CNS, such as the cerebral cortex, cerebellum, forebrain, and hindbrain[59]. While localization of PAX6+ cells has been within the TUJ1+ zone with defined

**Fig. 4 Cerebral organoid multi-layered organization. a** Immunofluorescence staining of cerebral organoid maturation markers (TUJ1/TBR1, NESTIN/TBR2, TUJ1/PAX6, N-CAD/FOXG1, CTIP2/SATB2) at 120 days of organoids matured in 4 dynamic systems, white arrows indicate neural rosette-like structures and yellow dashed lines indicate CP layer borders (scale bars = 100 μm for 10× and 5 μm for 25× magnification images, independent replicates = 3, Zeiss LSM 880). **b** A Schematic illustration of the cellular organization of matured cerebral organoids, MZ marginal zone, CP cortical plate, oSVZ outer subventricular zone, SVZ subventricular zone, VZ ventricular zone, oRG outer radial glial cell, vRG ventricular radial glial cell, IP intermediate progenitor cell. **c** WB analysis for specific maturation proteins of cerebral organoids matured in a shaker (1,5), spinner (2,6), RCCS (3,7), and μ-platform (4,8) at day 60 and 120, respectively (independent replicates = 2). **d** Heatmap data of the qRT-PCR analysis of cerebral organoids matured in static, spinner, shaker, RCCS, and μ-platform systems on day +30, day +60, and day +120 periods in comparison to generated organoids on day 15 (independent replicates = 2).

borders in RCCS organoids, regionally disorganized expressions of PAX6 were observed in a shaker, spinner, and μ-platform organoids with slightly decreasing protein levels toward day 120 (Fig. 4, Supplementary Fig. 1, Supplementary Fig. 2b, Supplementary Table 1). The adequate expression with the desired architecture of PAX6 expression suggests that RCCS may promote further neuronal differentiation in more mature cerebral organoids[51]. Apart from that, the upregulation of neural identity marker, SRY-box transcription factor 1 (SOX1), as well as downregulation of neural stem/progenitor cell marker, SRY-box transcription factor 2 (SOX2), indicate successful neural induction in cerebral organoids[6]. However, it is also known that the expression of SOX2 is upregulated again after neuronal and glial differentiation and provides the transcription of specific genes[60]. Immunofluorescence staining revealed no significant changes in SOX2 RNA levels (Fig. 4d, Supplementary Table 1) and SOX2+ cells (Supplementary Fig. 3) as the culture periods progressed but a decrease in SOX2 protein levels in μ-platform and RCCS organoids were noted (Fig. 4c, Supplementary Fig. 2b). Considering the cortical region, neurons are divided into two main groups: early born deep layer neurons and later born surface layer neurons. Chicken ovalbumin upstream promoter transcription factor integrating protein 2 (CTIP2) and special AT-rich sequence binding protein 2 (SATB2) genes are involved in the genetic control of these neurons, respectively[61,62], where the expressions dictate the spatial separation of cortical layers[63]. As such, deep-layer cortical neuron marker, TBR1 plays a significant role in brain development[64]. CTIP2, SATB2, and TBR1 were upregulated at protein and RNA expression levels in both RCCS and μ-platform organoids mainly on day 120 (Fig. 4, Supplementary Fig. 2b, Supplementary Fig. 3, Supplementary Table 1), showing an inside-out pattern of CP (yellow dashed lines indicate CP layer borders). Early-born neurons expressed CTIP2 and TBR1 in the deep layer, whereas late-born neurons expressed SATB2 in the superficial upper layer with the obvious spatial zone separation[65,66].

Besides, immunofluorescence staining revealed the presence of SATB2+ neurons at the outer/superficial regions, whereas CTIP2+ neurons populated at the inner regions of RCCS organoids, as well as the vast majority of TBR1+ cells similar to CTIP2+ cells at day 120, indicating the superiority of RCCS system to differentiate and mature brain-specific layers (Fig. 4a, b). Finally, to assess the expression pattern of neuron-specific apical epithelial cell surface marker, neural cadherin (N-CAD) and forebrain marker[54,67], forkhead box G1 (FOXG1), expressed early during cortical development[65,68], were examined with respect to both protein and RNA expression levels in all dynamic systems. Mostly, the RCCS system induced N-CAD expression in organoids as maturation duration increased, excluding the μ-platform. The expression level of FOXG1 is reported to increase at the neuronal differentiation to form the forebrain, remaining at the plateau[69]. Here, although FOXG1 was expressed in all dynamic systems throughout the whole maturation process, the expression levels suggested better stimulation of forebrain identity with cortical neuron's characteristic phenotype in μ-platform organoids at

day 120 (Fig. 4, Supplementary Fig. 2b, Supplementary Fig. 3, Supplementary Table 1). Besides, we examined zinc-finger transcription factor KROX20, a critical transcription factor in hindbrain development[70] and transthyretin (TTR), a main secretory protein of the choroid plexus epithelial cells that form the blood-brain barrier and produce the cerebrospinal fluid[71]. The expressions of KROX20 and TTR increased in all systems from day 30 in general, KROX20 was more expressed in RCCS organoids at day 60 and μ-platform organoids at day 120 (Fig. 4d, Supplementary Table 1). Therefore, laminar flow at certain levels of shear stress supported multi-layered organization, corticogenesis, early/late stage brain development, and formation of various regions in brain organoids.

**Laminar flow induces further maturation of cerebral organoids at the functional and molecular level.** The RCCS and μ-platform systems facilitated further maturation of cerebral organoids in terms of advanced maturation markers and transcriptome profiles. The expression of a mature synaptic marker, postsynaptic density protein 95 (PSD95) begins in the embryonic stage and continues in the postnatal period, playing a pivotal role in learning and memory formation, and is mostly related to forebrain and midbrain development[72,73]. Immunofluorescence staining and western blot analysis revealed that PSD95 was positively expressed mostly in RCCS organoids from day 60 to 120 (Fig. 5a–c, Supplementary Fig. 2c), as well as at RNA level in all systems as of 30 days in general (Fig. 4d, Supplementary Table 1). Interestingly, the PSD95+ layer also indicated a preplate splitting-like structure (indicated by a white dashed straight line with a two-sided arrow indicate) in RCCS organoids (Fig. 5a), as demonstrated by immunostaining against chondroitin sulfate proteoglycan in brain organoids at day 60[12]. Also, the expression of Prospero homeobox transcription factor (PROX1), an advanced key regulator for brain organogenesis as in early postnatal neurogenesis of the thalamus, the cerebellum, and the hippocampus, and also adult neurogenesis of the hippocampus related to midbrain and hindbrain[74] was upregulated in three dynamic system organoids as of day 60 (Fig. 5a–c, Supplementary Fig. 2c). The presynaptic vesicular glutamate transporter 1 (VGLUT1), an excitatory glutamatergic neuronal marker, which is known to emerge during the second trimester of the neonatal brain[75] was expressed at higher levels in both μ-platform and RCCS organoids at day 120 compared to the spinner system (Fig. 5a–c, Supplementary Fig. 2c) (since no expression was noted in organoids matured in the shaker). VGLUTs are responsible for the vesicular accumulation of glutamate, the major excitatory neurotransmitter that plays critical roles in neuronal signaling and cortical development, which is uploaded into synaptic vesicles within presynaptic terminals before undergoing regulated release at the synaptic cleft[76,77]. Therefore, we decided to examine the cumulative glutamate levels in one week of used maturation medium of RCCS and μ-platform organoids at days 30 and 60 to evaluate further maturation. Significant time-dependent increases were noted in glutamate levels for both groups ($p < 0.0001$ and $p < 0.01$, respectively), even more in RCCS organoids as

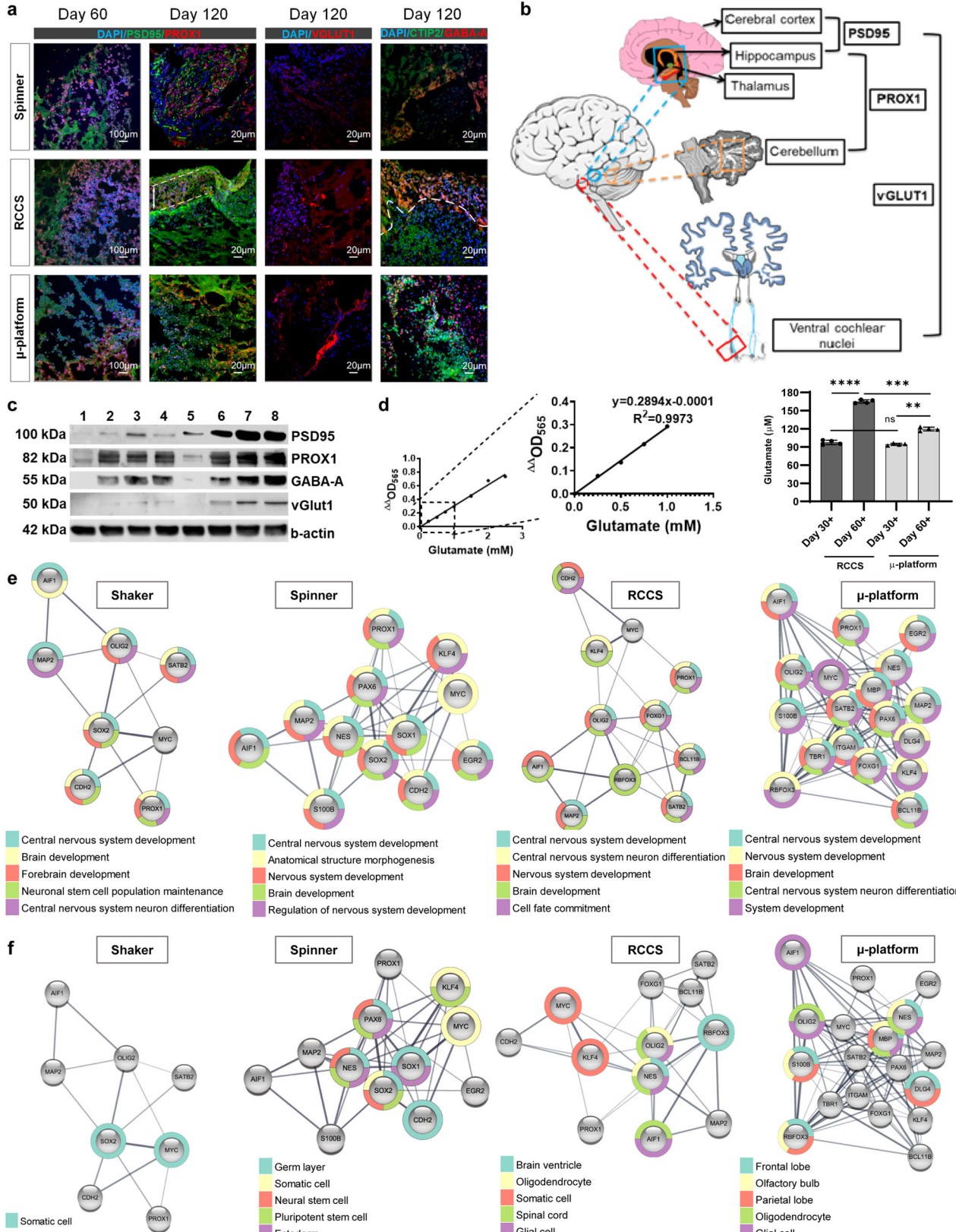

165 ± 2.8 μM ($p < 0.001$) (Fig. 5d, Supplementary Data 1c). During early corticogenesis, bursts of action potentials cause the spreading of giant waves of calcium influxes through the developing cortex, which is described as giant depolarizing potentials (GDPs), depending both on excitatory glutamate and gamma-aminobutyric acid (GABA) inputs. Glutamate-dependent GDP-like events were reported in neuronal organoids. Reduced GDPs in >40-day organoids with GABA polarity switch were regarded as indicators of progressive neuronal network maturation[78]. On the other hand, the glutamate level that was secreted from microglia was measured at around 20 μM in healthy in vitro neuronal cultures[79,80]. Given that, the change in the uptake and

**Fig. 5 Molecular and functional maturation of cerebral organoids. a** Immunofluorescence staining of advanced maturation markers (PSD95, PROX1, VGLUT1, CTIP2, and GABA-A) in cerebral organoids matured in dynamic systems on days 60 and 120, white dashed straight line with a two-sided arrow indicate preplate splitting-like structure, yellow dashed lines indicate CP layer-like border (scale bars = 100 µm for 10× and 20 µm for 25× magnification images, independent replicates = 3, Zeiss LSM 880). **b** Schematic illustration of functional matured brain regions. **c** WB analysis of PSD95, PROX1, GABA-A, and VGLUT1 proteins of cerebral organoids matured in a shaker (1,5), spinner (2,6), RCCS (3,7), and µ-platform (4,8) at days 60 and 120, respectively (independent replicates = 2). **d** Glutamate standard curve and extracellular glutamate concentrations (µM) in the supernatant of RCCS and µ-platform organoids on day 30 and 60 ($^{ns}p > 0.05$, $**p < 0.01$, $***p < 0.001$, $****p < 0.0001$, $n = 4$ for independent replicates = 2). **e** The top five biological processes (Gene Ontology) and **f** tissue expression (tissues) enrichments of upregulated genes (Log2 transformation of fold regulation ≥1.9, p-value < 0.05, independent replicates = 2) of cerebral organoids matured in dynamic systems on day 120.

consequently release rate of extracellular glutamate in 60-day RCCS and µ-platform organoids and growth in organoid sizes might be associated with more mature states. One of the distinguishing features of neuronal circuits and maturation is the switch from excitatory to inhibitory GABAergic neurotransmission. GABA-A, a presynaptic GABAergic interneuron receptor that modulates neurotransmitter release in both peripheral and central synapses[7,78,81–83], was more expressed in both RCCS and µ-platform organoids with CTIP2+ deep-layer of cortical neurons as of day 60 (Fig. 5a–c, Supplementary Fig. 2c). Thus, the neuronal network complexity was validated in RCCS and µ-platform organoids with the presence of both presynaptic glutamatergic VGLUT1+ and GABAergic GABA-A+ neurons, which are critically involved in neuronal network oscillations, as well as postsynaptic PSD95+ neurons, GFAP+/S100B+ astrocytes, and MBP+/OLIG2+ oligodendrocytes.

Herein, gene enrichment analysis was performed to identify the associated pathways of upregulated genes (Supplementary Data 2a) with respect to neural differentiation and organoid maturation. Based on biological process enrichment data, protein-protein-interaction (PPI) enrichment p-values of all days and systems were determined as $p < 0.05$. There was no significant enrichment (no correlation between upregulated genes) detected in organoids matured in the static system at days 30 and 60 and day 60 in µ-platform organoids. When the systems were compared based on maturation time, the highest enrichment numbers were determined at days 30 and 120 for the µ-platform and day 60 for the shaker, which enriched 82, 131, and 82 biological process terms, respectively (Supplementary Data 2b). *Regulation of neuron differentiation* (GO:0045664) was enriched in 11 different systems (different maturation systems with examined time intervals) and determined as the most enriched biological process, which was followed by the biological processes of *brain development* (GO:0007420), *cell fate commitment* (GO:0045165), *cell morphogenesis* (GO:0000902) and *central nervous system neuron differentiation* (GO:0021953), enriched in 10 different systems (Supplementary Data 2c).

The top 5 most enriched biological process terms for each dynamic system at day 120 were determined according to the false discovery rate (FDR) value. Among them, *central nervous system development* (GO:0007417) was the most enriched process for all systems, especially in the µ-platform (Fig. 5e). On the other hand, when tissue expression (TISSUE) enrichment of upregulated genes (Supplementary Data 3a) was examined, PPI enrichment p-values of all days and systems were determined as $p < 0.05$. There was no significant enrichment detected at day 30 for static and spinner systems, day 60 for the µ-platform, and day 120 for the static system. When the systems were compared based on maturation time, the highest enrichment numbers were determined at days 30 and 120 for µ-platform, and 60 for RCCS, which enriched 10, 25, and 35 tissue expression terms, respectively, and satisfactorily (Supplementary Data 3b). Although not thought to be very important in cerebral organoid maturation-related pathways and can only be associated with

stem cell differentiation, *the somatic cell* (BTO:0001268) term was enriched in 9 different systems and determined as the most enriched tissue expression term (Supplementary Data 3c), as in the shaker at day 120 (Fig. 5f). Besides, the 5 most enriched terms for each system were determined according to the FDR value. Among them, the highest enrichment numbers with respect to *the frontal lobe* (BTO:0000484), *oligodendrocyte* (BTO:0000962), and *glial cell* (BTO:0002606) terms, related to brain tissue, were achieved in µ-platform organoids, while brain ventricle (BTO:0001442) was the most related term to cerebral tissue, obtained in RCCS organoids at day 120 (Fig. 5f). Finally, when the protein expression and gene enrichment results are correlated, laminar fluid flow directed dynamic systems facilitated the functional and molecular maturation of neural populations in cerebral organoids even without exogenous neurotrophic factors such as dual SMAD inhibition and/or WNT signaling activation[51] in a relatively short period as of day 60.

**Laminar flow conditions improve the survival of brain organoids with reduced cell death, apoptotic zones, and adherens junctions by enhancing oxygen and nutrient transport.** We previously showed that cerebral organoids matured in RCCS and µ-platform expressed specific neuronal, glial, and endothelial cell markers providing cellular diversity, multi-layered functional neuronal–vascular organization, and ventricle-like structures in the cerebral cortex. Therefore, we hypothesized that the presence of controlled laminar fluid flow with lower shear stress would leverage brain organoid development by facilitating molecular nutrient-gas diffusion in and out of the organoids, leading to cell survival and further improvement of the neurodevelopmental process. Apart from FOXG1+ layers showing advanced neuronal differentiation, cells positive for vascular endothelial (CD31) and the proliferation (Ki-67) markers were abundant in cerebral organoids cultured in µ-platform (white arrows indicate neural rosette and neural tube-like structures) and RCCS, compared to the spinner and shaker systems in terms of both protein and gene expression levels, as maturation time has progressed (Fig. 6a, b, c, Supplementary Fig. 2d, Supplementary Table 1). An increase in the proliferative and vascular cell population along with reduced apoptosis in controlled laminar flow organoids resulted in noticeably longer survival rates than those in the higher shear stress organoids. It is known that shear stress generated by a flow in the system has a prominent effect on the proliferation and differentiation of cells. In laminar flow prevailing in physiological conditions where blood flow is mimicked, PSCs tend to differentiate into endothelial cells[84]. Endothelial cells are exposed to shear stress at a wide range in the vasculature[85], and sense the shear stress by various proteoglycans and glycosaminoglycans on their surfaces[86], resulting in biochemical responses such as the expression of specific endothelial markers. In our study, CD31 expression was observed to increase in endothelial cells that differentiated from iPSCs due to better mimicry of physiological conditions in low-shear stress systems with laminar flow. Vascularization in cerebral organoids was reported to upregulate the

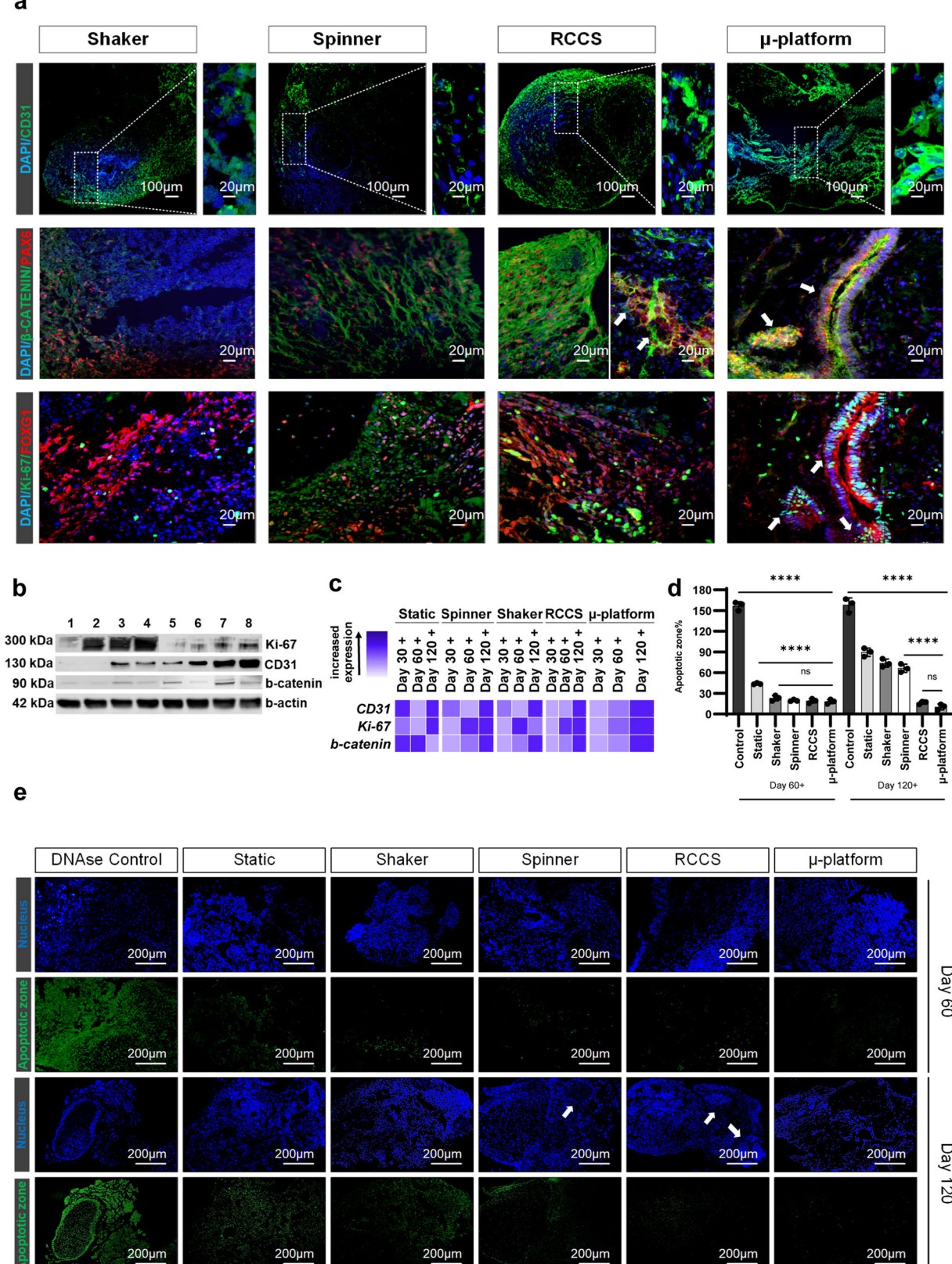

Wnt/β-catenin signaling and increase the Ki-67 cell proliferation marker. Endothelial-like cells in vascularized organoids signal to neural stem cells and regulate their self-renewal, and differentiation into neurons during CNS development[87]. Additionally, the adherens junctions control vRG cell's self-renewal, proliferation, differentiation, and survival via active Wnt/β-catenin/

N-cadherin signaling[88]. We also found that both protein and RNA expression levels of β-catenin were more elevated in microgravity-driven RCCS organoids and low shear stress-induced μ-platform organoids on day 120 (Fig. 6a–c, Supplementary Fig. 2d, Supplementary Table 1) as in N-CAD levels, while neural/glial cell diversity and CP-like structures were

**Fig. 6 Survival of cerebral organoids. a** Immunofluorescence stainings of cerebral organoids matured in 4 dynamic systems at day 120 with CD31, β-catenin, PAX6, Ki-67, and FOXG1 markers, white arrows indicate neural rosette and neural tube-like structures (scale bars = 100 μm for 10× and 20 μm for 25× magnification images, independent replicates = 3, Zeiss LSM 880). **b** WB analyses of Ki-67, CD31, and β-catenin proteins of cerebral organoids matured in a shaker (1,5), spinner (2,6), RCCS (3,7), and μ-platform (4,8) at days 60 and 120, respectively (independent replicates = 2). **c** Log2 transformation graph of the qRT-PCR analysis for CD31, Ki-67, and β-catenin gene expression of cerebral organoids matured in static, spinner, shaker, RCCS, and μ-platform systems on day +30, day +60, and day +120 periods in comparison to generated organoids on day 15 (independent replicates = 2). **d** Percentage of the apoptotic zone ($n = 3$ for independent replicates = 3, $^{ns}p > 0.05$, $^{****}p < 0.0001$) and **e** TUNEL apoptosis fluorescent staining images of sectioned organoids at days 60 and 120. DNAse-treated organoid section matured in the RCCS system is used as a control group. The DAPI-stained cell nucleus (blue), green fluorescent stained apoptotic zones in the cell nucleus (green), and white arrows indicate neural rosette-like structures (scale bars = 200 μm, independent replicates = 3, Zeiss Axio Vert.A1).

observed (white arrows indicate PAX6+ neural rosette and neural tube-like structures).

During the organoid maturation process, TUNEL fluorescent stainings, where cell viability, cell death, and lifespan could be examined[89], were performed to determine qualitatively and quantitatively possible apoptotic regions in organoids harvested at days 60 and 120 from all dynamic systems and compared with that static culture (Fig. 6d, e, Supplementary Data 1d). The ability of laminar flow systems to provide oxygen supply to the core regions of the organoids associated with cell survival was assessed as well. The cell nuclei of the DNAse-treated RCCS group stained with DAPI showed absorbance of green fluorescence, indicating that the control group was functioning successfully, and cell apoptosis was characterized by labeled fragmented DNA breaks. Labeling occurs by catalytic incorporation of fluorescein-12-dUTP into the 3′-OH end of DNA by the recombinant deoxynucleotidyl transferase enzyme (rTdT)[90]. When all maturation systems were examined at day 60, almost no elevated rates of DNA breakage and cell apoptosis were observed, while the significantly highest ($p < 0.0001$) cell death was observed in organoid sections matured in static culture via TUNEL+/DAPI+ cells at the apoptotic zone of 44.1% ± 1.1, compared to control group (157% ± 6.2). Among dynamic systems examined, although not statistically significant ($p > 0.05$), the highest DNA breakage was observed in the shaker, where the organoids were exposed to the highest shear stress, whereas the lowest apoptotic signals were noted in μ-platform and RCCS organoids exposed to at least 100-fold less shear stress. Approaching day 120, the presence of apoptotic cells in the inner regions of the organoids cultured under static conditions dramatically increased to 89.6% ± 6.2 due to nutrient-oxygen deficiencies, as the organoid size increased[91]. Besides, the cellular damages in the peripheral regions of organoids matured in shaker and spinner cultures were observed to increase from day 60 to 120 ($p < 0.0001$, apoptotic zone of 73.8% ± 5.9 and 66.5% ± 6) due to uneven hemodynamic forces and higher shear stresses. On the other hand, the detection of few apoptotic cells in organoids cultured in μ-platform and RCCS yielded the highest survival rates with significantly lower apoptotic zones of 10.4% ± 4.2 and 16.5% ± 2.2, respectively. Consequently, organoids matured in dynamic systems showed more prominent cell viabilities by reduced cell deaths at the center of organoids with sufficient nutrient-gas supply and less cell apoptosis-induced organoid survival due to the effective agitation in different dynamic flow regimes compared to the static group[6,92], mostly in the μ-platform[12,93] and RCCS systems that provided complex cytoarchitecture in organoid maturation.

## Discussion

Compared to traditional mouse models and monolayer neuronal cell cultures, the most advanced technology for modeling human brain development used to date is human brain organoids, first developed by Lancaster et al.[6], then improved by different groups over the years by integrating bioengineering approaches such as usage of biomaterials[94], chemical additions[10,95], genetic

variations[91], and utilizing dynamic systems[12,20,23,26,31]. The cerebral organoids recapitulate "unique human" features of brain development such as the presence of the outer SVZ, rich in the neurogenic outer radial glia based on the cortical layer, various brain regions identities like forebrain and hindbrain, and also gene expression profiles and electrophysiological properties. Thus, information about human-specific brain development can be obtained in the most appropriate genetic context. Molecularly and functionally matured cerebral organoids are used as state-of-the-art technology to study human brain development, model neurodevelopmental-neurodegenerative conditions, and study preclinical drug-gene interactions for potential therapies. Significant efforts have been devoted to comprehending how fluid flow and the physical forces affect embryonic development[96,97]. Herein, we developed two processes exposed to dynamic flow, leveraging the maturation of cerebral organoids. The flow regime was laminar, where adjacent layers of cell culture medium slide along one another reducing the friction on the inner walls of RCCS STLVs and μ-platform. Indeed, the maximum shear stress values of $7.72 \times 10^{-3}$ and $2.32 \times 10^{-4}$ Pa were achieved, respectively. However, much higher shear stress values of 1.57 and 0.96 Pa were computed for the shaker and spinner by using COMSOL Multiphysics. Physical forces, such as shear stress, gravity, and cyclic stretch affect mechanosensitive pathways or change the expression levels[22]. Recently, the transcriptomic analyses of cerebral organoids showed that mechanotransduction-associated genes including integrins, β-catenin, Wnt, and Delta-like pathway genes change under physical stress conditions[98]. Thus, organoids exposed to varying magnitudes of hydrodynamic forces are expected to display differences in structural and functional features during development. As such, cell proliferation of hPSCs has been shown to increase under simulated microgravity culture conditions with significantly higher expression levels of Ki-67 compared to 1 g culture condition, leading to enhanced self-renewal, as revealed by the increased protein levels of the core set of pluripotent transcription factors[99]. In another study, human embryonic stem cell (hESC)-derived forebrain-specific neural-cortical organoids were shown to have the highest expression of Ki-67 when cultured under RCCS conditions after day 14 of the generation process compared to organoids cultured in static and other days of the generation in RCCS[31]. These results are in agreement with our study and support the increased expression levels of Ki-67 in RCCS organoids. Also, the use of RCCS under microgravity conditions in embryonic body formation from ESCs has been shown to promote more homogeneous EB formation and endoderm differentiation by modulating the Wnt/β-catenin pathway[100]. On the other hand, the most severe subtype of spina bifida, which is one of the neural tube defects that starts in the 4th week of pregnancy, occurs when the spinal cord is exposed to shear stress from amniotic fluid[101,102]. This negative effect of shear stress on the brain and spinal cord formation during embryonic development highlights the importance of shear stress during organoid maturation. RCCS and μ-platform organoids

have reached ideal sizes, approximately 95% harvestability, prolonged culture time with Ki-67+/CD31+/β-catenin+ proliferative, adhesive, and endothelial-like cells and exhibited (1) enriched cellular diversity (abundant neural/glial/ endothelial cell population), (2) structural brain morphogenesis (complex CP, SVZ, and VZ-like layers, radial–glial organization, preplate splitting), (3) further functional neuronal identities (glutamate secreting glutamatergic, GABAergic and hippocampal neurons) and synaptogenesis (presynaptic-postsynaptic interaction) during whole human brain development. GFAP+/S100B+ astrocytes and NEUN+/MAP2+ mature neurons were abundant on day 60 in RCCS organoids richest in cell diversity, both RCCS and μ-platform organoids expressed CD11b+/IBA1+ microglia and MBP+/OLIG2+ oligodendrocytes at high levels as of day 60. The potential role of microglia as resident macrophages of the central nervous system (CNS) in neuroinflammation has implications in neurodegenerative diseases such as stroke, traumatic brain injury, Alzheimer's disease, Parkinson's disease, and amyotrophic lateral sclerosis, and also neurodevelopmental /neuropsychiatric disorders including schizophrenia, autism spectrum disorder, addiction, and depression. A combined microglial dysfunction and activation of pro-inflammatory responses are associated with the etiology of pathogenesis[103–106]. In this context, the use of 60 days-old RCCS and μ-platform cerebral organoids will be advantageous for in vitro modeling of mentioned neural diseases, where microglia play a significant role, as well as other mature neural and glial cells. RCCS and μ-platform organoids at day 120 contained CD31+ endothelial-like cells and GFAP+/CD11b+ glial cells. Neural stem cells and endothelial cells interact through a signaling cascade, involving autocrine and paracrine factors[107] and allow basement membrane interactions with other cells, which could enhance organoid maturation[108]. Analogical reasoning might be considered especially for μ-platform organoids, where the upregulation of CD31 was remarkable and the highest enrichment numbers with respect to the *frontal lobe* (BTO:0000484), *oligodendrocyte* (BTO:0000962), and *glial cell* (BTO:0002606) terms, related to brain tissue and *central nervous system development* (GO:0007417), being the most enriched process were elicited as the μ-platform. On the other hand, the presence of endothelial cells might allow the blood–brain barrier (BBB) associated neurological disorders such as neuroAIDS, COVID-19, and multiple sclerosis to be studied as a model for molecular trafficking of potential drugs. The multicellular BBB organoids provide data about humanized functional complexity and vascular-like structures comparable to in vivo studies[109,110]. RCCS organoids displayed CP, SVZ, and VZ architectures at day 120, where TUJ+ cells were located at the basal side of the ventricle-like structures and TBR1+ cells were in the deep layer of cortical-like plates. Well-defined progenitor zone organization, neural identity, and further neuronal differentiation with TBR2+/PAX6+/NESTIN+/FOXG1+ and SOX2+/SOX1+ cells (white arrows indicate neural rosette-like structures) were observed as well. Similarly, these markers were highly expressed in μ-platform organoids at both protein and gene expression levels (white arrows indicate PAX6+ and FOXG1+ neural rosette and tube-like structures). Considering the maturation of brain-specific layers, the later-born upper-surface layer SATB2+ neurons at the outer/superficial regions of RCCS organoids were notable, as well as the early-born deep-layer CTIP2+/TBR1+ neurons at the inner regions at day 120 (yellow dashed line indicates CP layer-like border). Likewise, CTIP2+, SATB2+, and TBR1+ cells were also abundant throughout the CP in μ-platform organoids mainly at day 120. On the other hand, microgravity driven β-catenin+ adherens, N-CAD+ apical epithelial, and TTR+ choroid plexus epithelial cells were populated in RCCS organoids as of day 60, while both FOXG1+ forebrain, KROX20+ hindbrain and

PROX1+ hippocampus related to midbrain and hindbrain identities were observed in μ-platform organoids at day 120. Additionally, both RCCS and μ-platform organoids exhibited advanced maturation with both presynaptic VGLUT1+ glutamatergic, GABA-A+ GABAergic, and postsynaptic PSD95+ neuronal cells at day 120. Overall, a functional preclinical 3D in vitro brain model has been created, recapitulating the early/late brain development processes with RCCS and μ-platform organoids that expressed all these maturation-related markers in advance at different stages during the cerebral organoid formation process. The neurotoxic effects of environmental and genetic factors on different stages of brain development[23,111–113], microbial-viral infections in the CNS[114], potential drug molecules, and gene therapies can be tested by highly functional RCCS and μ-platform organoids. While the engineering of physiologically matured cerebral organoids advances, scaling up will be an issue to resolve and accelerate the adoption of the technology to the pharmaceutical industry. Herein, approximately 100 functionally matured cerebral organoids have been produced in 55 ml RCCS STLVs with >95% harvestability. Although scale-up was not planned for this study, we presume that about 1000 matured organoids can be harvested in one batch, if we scale up to 500 ml RCCS STLVs, as such the μ-platform can be multiplied by connecting in a parallel or serial manner for scale-up purposes. This study suggests that RCCS and μ-platform organoids showing a high level of physiological fidelity can serve as functional preclinical models for testing new therapeutic regimens and benefit from multiplexing.

## Methods

**Maintenance of human induced pluripotent stem cells (iPSCs)**. iPSC lines that were previously reprogrammed from human dermal fibroblasts of healthy donors and characterized in terms of pluripotency markers and mycoplasma purity, were obtained from Izmir Biomedicine and Genome Center, Stem Cell and Organoid Technologies Laboratory[115]. Well-shaped iPSC colonies were manually picked, passaged with 1 mg/mL dispase solution (Stem Cell Technologies), and expanded on hESC-qualified Matrigel matrix basement membrane (Corning) with mTeSR1 medium (Stem Cell Technologies) with the daily change until differentiation.

**Generation of cerebral organoids**. The cerebral organoid generation steps were based on the detailed method of Lancaster and Knoblich[116] with some minor modifications regarding the durations in consecutive steps starting from maturation, in which dynamic systems and flow rates are different from the applied method.

*Embryoid body formation and differentiation of germ layer*. When iPSCs attained a confluence of 70–80%, the cells were suspended as single cells with gentle cell dissociation reagent (Stem Cell Technologies), resuspended in hESC media [DMEM/F12 medium (Gibco) supplemented with 20% KOSR (Gibco), 3% hESC-qualified FBS (Gibco), 1% Glutamax (Sigma-Aldrich), 1% NEAA (Sigma-Aldrich), 1% Penicillin/Streptomycin (Sigma-Aldrich), 0.007% BME (Sigma-Aldrich)] with 4 µg/mL bFGF (Peprotech) and 50 µM Y-27632 (Tocris) and cultured to obtain EBs and ectoderm differentiation at 9000 live cells/150 µL in a low attachment 96-well U-bottom plate (Corning) at 37 °C and 5% CO₂. The hESC media was changed by half every other day without disturbing the EBs at the bottom of the well. EBs were observed microscopically for about 5-6 days until they started to brighten and reached diameters >350–400 µm.

*Induction of neuroepithelial structure*. When EBs were about 500–600 µm in diameter, started to brighten and had smooth edges, 1–4 EBs were transferred gently with a cut 200 µL pipette tip to one well of a low attachment 24-well plate containing 500 µL neural induction media [(DMEM/F12 medium with 1% N2 supplement (Gibco), 1% Glutamax, 1% MEM-NEAA, 1% penicillin/streptomycin, and 1 µg/mL Heparin (Stem Cell Technologies)]. EBs were monitored for additional 4–5 days with a medium change every other day until they formed brighter structures around the edges indicating neuroectodermal differentiation. Once these regions began to show radial organization of a pseudostratified epithelium consistent with neuroepithelium formation, EBs were ready for the next step.

**Transfer of neuroepithelial tissue to matrigel drops and stationary culture of expanding neuroepithelial buds**. After the neuroepithelial tissue were formed,

each tissue was carefully transferred with a cut 1000 µL pipette tip to each micro concave structure formed on sterile parafilm, excess liquid was drawn and 25 µL liquefied matrigel (Corning) at +4 °C was added on tissues for 1–2 h. A quick pipette tip was used to position the tissues in the center of the matrigel drops and remove air bubbles. After waiting for about half an hour at 37 °C for the matrigel drops to polymerize, the solidified drops were transferred to 6 well suspension culture plates in a cerebral organoid differentiation medium [1:1 ratio of DMEM-F12:neurobasal media supplemented with 0.5% N2 supplement, 0.025% Insulin (Sigma-Aldrich), 1% Glutamax, 0.5% MEM-NEAA, 1% penicillin–streptomycin, 0.035% BME (1:100 in medium), and 1% B27 supplement without vitamin A (Gibco)]. The embedded tissues were cultured for 3–4 days until they began to form more enlarged neuroepithelial buds containing fluid-filled cavities, with a medium change every other day. After this process, the maturation of cerebral organoid structures whose generation was completed, was continued in dynamic culture as well as in static culture, in cerebral organoid differentiation medium containing vitamin A.

**Maturation of cerebral organoids under dynamic conditions**. Organoid maturation processes with four different dynamic flow regimes (created by spinner, shaker, µ-platform and RCCS bioreactor) was carried out to promote self-neuronal organization, efficient oxygen and nutrient delivery to the tissue, and also ensure the formation of larger brain organoids with fluid-filled cavities resembling ventricular spaces and cerebral cortex structure. Various flow rates and shear stresses, formed under different flow conditions were calculated and numerically simulated by COMSOL Multiphysics to mimic hemodynamic forces on organoids and increase in vivo like structural development and transcriptional outputs.

*Computational fluid dynamics (CFD) simulation*. CFD simulations were performed for determining the shear stresses under various flow conditions and compared novel organoid generation methods (µ-platform and RCCS) with those of conventional processes (spinner and shaker) using finite element commercial code in COMSOL Multiphysics. For simulation of shear stress, single-phase flow under laminar conditions following the Navier-Stokes equation was applied to µ-platform and RCCS. Also, turbulent flow conditions following the k-ε equation were applied to spinner and shaker. All the equations and variables used for the computational models are listed in Supplementary Fig. 4.

*Maturation in shaking bioreactor*. Approximately, 40 of the generated cerebral organoids with more enlarged neuroepithelial buds containing fluid-filled cavities were transferred to a sterile glass erlenmeyer flask with a working volume of 30 mL at day 15 and were cultured in cerebral organoid differentiation medium containing vitamin A on a shaking agitated culture system (Grant POS-300) operated at 100 RPM (increased to 110 RPM as organoids grow), in an incubator at 37 °C and 5% CO$_2$[6,11,92]. The culture medium (¾ volume) was refreshed twice a week, and the culture was maintained for 120 days.

*Maturation in spinning bioreactor*. The generated cerebral organoids (40 pieces) were transferred to a sterile spinner, the smallest stirred tank bioreactor with a working volume of 30 mL at day 15[6,10]. Dynamic maturation was performed in cerebral organoid differentiation medium containing vitamin A, on a magnetic stirrer (DLAB, MS-C-S4) operated at 50 RPM (specially designed for cell culture studies with reduced titration, allowing organoids to disperse homogeneously without physical degradation and aggregation, increased to 60 RPM as organoids grow), in an incubator at 37 °C and 5% CO$_2$. The ¾ volume of the growth medium was changed twice a week, microscopic/macroscopic observations were made regularly, samples were collected for characterization tests at days 30, 60, and 120 and the culture was maintained for 120 days.

*Maturation in rotary cell culture system (RCCS) bioreactor*. The generated cerebral organoids (40 pieces) were seeded in 55 mL slow turning lateral vessels (STLVs) for a dynamic microgravity driven maturation. The volume was adjusted to completely fill the vessel with a cerebral organoid differentiation medium containing vitamin A and all air bubbles were removed by syringes. STLVs were rotated on the RCCS-1 system (Synthecon Inc.) with a starting rotation of 10 RPM (increased to 15 RPM as organoids grow) in an incubator at 37 °C and under a 5% CO$_2$ and 95% air atmosphere. Rotation speed was increased by 1 RPM when needed to prevent organoids from collapsing. The maturation medium (¾ volume) was refreshed once a week, air bubbles were also removed as needed over the culture period and culture was maintained for 120 days.

*Maturation in µ-platform*. The developed organoid-on-chip platform that can be easily assembled and disassembled, was created using Polydimethylsiloxane (PDMS, SYLGARD 184) which is a bioinert and biocompatible polymer, for mimicking the in vivo vascular system and cerebral structure of the brain. The size of the platform and 5 hemispherical wells were designed in 3D CAD program (SolidWorks) with 12 mm diameters. PDMS solution was prepared by mixing PDMS base elastomer with curing agent at a ratio of 10:1, and a homogeneous and transparent structure was obtained by removing air bubbles from the solution with a vacuum pump (EduScience). Subsequently, homogeneous PMDS solution was

poured into the mold, which was obtained using a 3D stereolithography printer and cured for 6 h at 50 °C. Sterilization of the platform was completed by exposure to ethylene oxide gas for 16 h. For maturation, 5 of the generated cerebral organoids were transferred to each hemispherical well of the µ-platform, clamped by a pair of polymethylmethacrylate (PMMA) sheets, tightened with screws and bolts to ensure hydraulic tightness. The circulation of the maturation medium was provided from the inlets and outlets from both sides of the µ-platform, by the reservoir integrated into the dynamic flow system with a peristaltic pump (Longer pump, BT100-2J) and kept at the incubator at 37 °C and 5% CO$_2$.

## Characterization of matured organoids

*Immunostaining assay*. Matrigel was removed by washing with cold 1× PBS in the canonical 15 ml tube. Organoids were then fixed with freshly prepared 4% paraformaldehyde (PFA, Sigma-Aldrich) on ice for 30 min. The fixated organoids were washed three times with 1× PBS. Following fixation, organoid staining was performed either by frozen sections or paraffin embedded sections. *To prepare cryo frozen sections*, organoids were first embedded in the mold comprising OCT matrix (Tissue-Tek) at −20 °C and serial cryo-sections from specimens were obtained at 10–20 µm thicknesses. *To prepare paraffin embedded sections*, dehydration of organoids was performed by applying different concentrations of ethanol, then samples were embedded in paraffin (Sigma-Aldrich) and 5 µm thick sections were used for the staining experiments.

For immunofluorescence (IF) staining, cryo-frozen sections were kept at room temperature (RT) for 15 min, permeabilized with 0.5% Triton X-100 (depending on the protein localization) for 15 min at RT and blocked with 1% FBS, 0.5% BSA, 1.15% glycine and 0.1% Tween-20 in 1× PBS for 1 h at RT. Then, primary antibodies (Supplementary Table 2) were diluted in the blocking buffer and samples were incubated for 24 h at 4 °C. Subsequently, samples were washed three times with 1× PBS and Alexa fluor-conjugated secondary antibodies (Supplementary Table 2) were diluted in a blocking buffer and incubated for 1 h at RT. To visualize the cell's nucleus, DAPI-staining (Sigma-Aldrich) was performed and imaged by confocal microscopy Zeiss LSM 880.

For immunohistochemistry (IHC) staining analysis, paraffin embedded sections were dewaxed in xylene (Sigma-Aldrich) and rehydrated in decreasing concentrations of ethanol. Afterwards, sections were stained with Hematoxylin (BioOptica) for 10 min at RT, stained with Eosin (BioOptica) for 1 min at RT, incubated in an alcohol series and xylene, then mounted with entellan (Merck). Finally, sections were evaluated under light microscopy (Zeiss Axio Vert.A1).

*TUNEL apoptosis test*. DeadEnd™ Fluorometric TUNEL System (Promega, G3250) assay was used to examine apoptotic regions in matured organoids. Initially, the paraffin embedded sections of fixed organoids were washed twice in 1X PBS for 5 min and permeabilized in 0.2% Triton® X-100 in PBS for 5 min. After rewashing, sections were equilibrated with a 100 µL equilibration buffer at RT for 10 min. Then, sections were labeled with 50 µL of TdT reaction mix and incubated for 60 min at 37 °C in a humidified chamber in the dark. To stop the reaction, sections were washed in 2× SSC for 15 min and three times in PBS for 5 min. Finally, DAPI-staining was made to visualize all nuclei in tissue for 10 min at RT, sections were mounted with mounting medium and the localized green fluorescence of apoptotic cells in tissue were detected by fluorescence microscopy (Zeiss, Axio Vert.A1). We used the ImageJ program to quantify apoptotic zone vs nucleus and data were then calculated with GraphPad Prism 8.3.0.

*Western blot (WB) assay*. Organoid lysates were harvested using RIPA lysis buffer (Pierce, Thermo Scientific) including protease-phosphatase inhibitor cocktail (Cell Signaling, 5872). Then, the amount of total protein in lysates was determined by Bicinchoninic acid (Pierce™BCA Protein Assay Kit, 23225, Thermo Scientific) assay. Lysates were separated on a 10% sodium dodecyl sulfate-polyacrylamide gel (SDS-PAGE) gel with protein standard (Bio-Rad, 161-0394) and then transferred to a PVDF membrane (Thermo Scientific). Blots were incubated with primary antibodies and anti-beta Actin antibody as a 'housekeeping' protein (Supplementary Table 2) in blocking buffer containing 5% skim milk overnight on rocker shaker (Heidolph) at +4 °C. Then blots were probed with HorseRadish Peroxidase-conjugated secondary antibodies (Supplementary Table 2) for 1 h at RT. Next, the solutions in the Clarity Western ECL Substrate (Bio-Rad) kit were mixed 1:1 and applied to the blot. The light formed at the end of the chemical reaction was measured with a chemiluminescence imaging system containing a CCD camera (Bio-Rad, ChemiDoc MP Imaging System) at a wavelength of 428 nm.

*qRT-PCR analysis*. Total RNA isolation from cerebral organoids was performed by using RNeasy Plus Mini Kit (Qiagen, Germany) and complementary DNA was synthesized for each group via RT2 First Strand Kit (Qiagen, Germany). Gene expression levels were determined by real-time PCR method using RT$^2$ SYBR Green qPCR Mastermix and gene specific primers (Supplementary Table 3) via LightCycler 480 Instrument II (Roche, Germany). The gene expression level of each gene was normalized using *HPRT1*, *GAPDH*, and *ACTB* housekeeping genes and Log2 transformation was calculated using the $2^{-\Delta\Delta Ct}$ method (Supp. Table 1). The normalized RNA expression levels of the specific neuronal/glial cell markers in different cell types (microglias, oligodendrocytes, astrocytes and neurons) downloaded from RNA single cell type data (Human protein atlas database; https://www.proteinatlas.org/about/download; rna_single_cell_type.tsv.zip; Karlsson et al., 2021). Also, Biological Process

(Gene Ontology) and Tissue expression (TISSUES) enrichments of upregulated genes (Log2 transformation of fold regulation ≥ 1.9, $p$ value < 0.05) of each system and day groups were realized via STRING v11.5 database (https://string-db.org/; Szklarczyk et al., 2021). The minimum required interaction score was identified as medium confidence (0.400), and the significant false discovery rate (FDR) was accepted as $p < 0.05$. Interactions were illustrated by using Cytoscape 3.8.2 software.

*Fluorescence-activated cell sorting (FACS) of microglia and astrocytes from organoids.* Initially, for single-cell suspension from RCCS organoids, three organoids were freshly harvested from vessels, collected to tube, washed with 1× PBS and immersed in 1 mL DMEM/F12 included papain (18.6 U/mL, P3125, Sigma-Aldrich) and DNAse 1 (337 U/mL, EN0521, Thermo Fisher) at 37 °C for 30 min on shaker. It was mechanically pipetted and vortexed at 10-min intervals under sterile conditions. Next, 2% FBS was added to stop the enzymatic reaction and the resulting single-cell suspension was centrifuged at 400 rfc for 5 min. An additional incubation was made in PBS buffer (included 2 mM EDTA, 1% FBS and 337 U/mL DNAse 1, pH 7.4) at RT for 15 min, and single-cells were passed through a 70 μm cell-strainer. Then, single-cells were labeled with CD45 and CD11b primary antibodies for microglia sorting and were labeled with GFAP primary antibody for astrocyte cell sorting (Supplementary Table 2) in FACS buffer (filtered 1× PBS included 2% BSA, 50 mM EDTA, and 5 ng/mL SCF, pH 7.4) at RT for 1 h. Finally, after centrifugation and 1× PBS washing, single-cells were labeled with Alexa fluor-conjugated secondary antibodies (Supplementary Table 2) in the FACS buffer and incubated for 1 h at RT. To detect the dead cells, DAPI-staining was performed after 1xPBS washing and cells were sorted/gated that were alive, single, and CD11b +/CD45+ or GFAP+ by the FACSAria III with 100 μm nozzle tip[42,43,117].

*Glutamate analysis.* To ascertain advanced maturation of RCCS and μ-platform organoids, we examined the release of glutamate. For this, samples of the medium (one week of used) were collected on day 60 of maturation process and analyzed with a colorimetric glutamate assay kit (MAK330-1K, Sigma-Aldrich), according to the manufacturer's instructions. Samples of fresh cerebral organoid differentiation medium were also run as negative controls.

*Statistics and reproducibility.* Each dynamic system has been run with two independent replicates of different iPSCs passages at two different times. The harvestability % (considering the culture batches $n = 2$ for independent replicates $= 2$), organoid size in mm (totally $n = 40, 33, 20, 10$ for static, n = 40, 40, 34, 24 for shaker, $n = 40, 36, 16, 8$ for spinner, $n = 40, 40, 30, 18$ for RCCS and $n = 40, 38, 28, 18$ for μ-platform, all for on day 0, 30, 60, 120, respectively), glutamate level as μM (considering the culture batches $n = 4$ for independent replicates $= 2$) and apoptotic zone % (considering the individual organoids $n = 3$ for independent replicates $= 3$) data were statistically analyzed by one-way ANOVA Tukey or Dunnett's multiple comparisons test with ±95% confidence interval and $p$ values < 0.05 were considered statistically significant in GraphPad Prism 8.3.0 program. Data were presented as the mean ± standard deviation with individual data points. In order to indicate degree of significance, $^{ns}p > 0.05$, $^*p < 0.05$, $^{**}p < 0.01$, $^{***}p < 0.001$, $^{****}p < 0.0001$ were used. The Student's $t$-test was also used for qRT-PCR analyses (independent replicates = 2 with RNA isolate pool containing at least two organoids from the same batch). On the other hand, H&E, IF and TUNEL staining were made with three independent replicates using three individual organoids at different times, while WB analysis were carried out with two independent replicates using protein lysate pool from at least two organoids from the same batch.

*Reporting summary.* Further information on research design is available in the Nature Portfolio Reporting Summary linked to this article.

## Data availability

Source data for the presented figures are provided as Supplementary Data 1–3 with this paper. Further simulation data that is generated and analyzed during the current study is available from the corresponding author on reasonable request. The normalized RNA expression levels of the specific neuronal/glial cell markers in different cell types are downloaded from RNA single cell type data from Human protein atlas database; https://www.proteinatlas.org/about/download; rna_single_cell_type.tsv.zip. In addition, Biological Process (Gene Ontology) and Tissue expression (TISSUES) enrichments of upregulated genes of each system and day groups are realized via STRING v11.5 database (https://string-db.org/).

## Code availability

Details of publicly available software used in the study are given in the "Methods and Data availability" section. Apart from this, no special custom code or mathematical algorithms were central to reaching the conclusions of this work.

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

## Acknowledgements

Financial support provided by The Scientific and Technological Research Council of Turkey (TUBITAK) under grant number 119M578 is highly appreciated. In addition, P.S.M. gratefully acknowledges the TUBITAK 2211-A National Graduate Scholarship Program.

## Author contributions

P.S.-M. designed all the experiments, analyzed the data and wrote the paper. U.D. designed and fabricated the μ-platform. Y.F., G.B., and B.Y. performed maturation experiments. S.A. performed iPSCs cultures, established brain organoids, prepared samples and performed immunofluorescence staining and confocal microscopy with P.S.-M. Also, Y.F. assisted immunohistochemistry stainings and prepared samples for qRT-PCR. C.B.-A. designed primers for qRT-PCR and B.G. performed the qRT-PCR analysis. S.Y. performed computational fluid dynamics simulations. E.E. provided advice on data interpretation. O.Y.-C. conceived the project, supervised all the experiments and edited the whole paper.

## Competing interests

The authors declare no competing interests.
