## [Peer Review File · Communications Biology]

Reviewers' comments:

Reviewer #1 (Remarks to the Author):

In this manuscript, Saglam-Metiner et al. describe two novel engineering approaches (RCCS and u-platform) with the goal to improve harvestability, scalability, reproducibility and survival of human brain organoids during long term cultures. The authors start by elegantly demonstrating that the proposed models (RCCS and u-platform) exhibit less shear stress when compared to traditional cultures on shakers or spinner flasks. Moreover, the shear stress distribution in RCCS and u-platform is a lot more homogeneous than when compared to the current available systems (shakers and spinners). These novel methods offer a great potential to further improve quality and durability of human brain organoids in vitro, which have been negatively impacted by the limitations associated with current culture conditions. Despite the great potential, the manuscript needs significant improvement on the characterization of differentiation and maturation of cells within the human brain organoids to conclusively assess the value of these approaches to advance the field.

Major concerns:

1. Cell composition variability is known to be a limitation of some human brain organoid protocols. In this study, the authors refer to reproducibility in terms of organoid size, which does not infer much about the quality of the differentiation and cell composition. Therefore, conclusions about reproducibility should take other parameters in consideration, such as cellular composition and cellular organization, instead of just organoid size.

2. Overall presentation and description of data should be significantly improved. There is insufficient characterization that limits interpretation and conclusions. Moreover, proportion of cell types and protein expression patterns present in the study are inconsistent with what is known in the field. This data should be revised and validated more carefully. Below are some examples that require careful revision:

a) Brightfield images (Fig. 2) and immunostainings throughout the manuscript raise concerns about the organoid differentiation quality. It is very hard to assess the quality of differentiation by these images and it is not very clear what information the authors want to convey with the H&E staining (Fig. 2). Additionally, the transparent regions observed in organoids from most conditions resemble cystic formation associated with off-target differentiation. Therefore, the authors should demonstrate that the organoids are indeed cortical organoids. This can be shown by high-quality images demonstrating a high proportion of FOXG1+ cells. FOXG1 is a forebrain marker that is expressed early during cortical development (PMID: 18983967, PMID: 24277810, PMID: 23995685). Moreover, the authors should include high-quality images including other standard markers to confirm successful cortical differentiation (e.g. EMX1, PAX6, SOX2, TBR1, CTIP2, SATB2), as well as verify the presence of rosette and neural tube-like structures in multiple timepoints.

b) The authors claim that both astrocytes (GFAP+) and neurons (NEUN+) are present since day 60 of organoid development. Previous studies with human brain organoids have demonstrated that differentiation in these in vitro 3D models recapitulate endogenous development where gliogenesis happens after neurogenesis. In one of these papers, astroglia are not seen before 6 months of development in vitro (PMID: 28445462). It is possible that modifications done by the authors to the protocol change this pattern, but it is recommended that higher magnification images, including more markers per cell type, are included to better illustrate the presence of these cells in their models.

c) The authors claim presence of cell types such as microglia and oligodendrocytes that are usually not present in most protocols. It is also recommended to include higher magnification images of these cells with more markers (microglia: IBA1, P2RY12, TMEM119; oligodendrocytes: MBP, OLIG2, PLP) in order to validate their presence in the organoids.

d) Staining for the postsynaptic marker, PSD95, seems unspecific in the representative images and would require co-staining with a presynaptic marker and imaging using a confocal microscope.

e) KI67 staining should be nuclear; however in Fig. 6a the staining seems to be outside of the nucleus. Moreover, specifically in this panel, representative organoids depicted in the RCCS and μ -platform raise concern about the quality of differentiation, as broad morphology seems cystic and irregular.

f) Font size in Fig. 1 should be increased in order to be seen.

g) Other examples within Fig. 3:

i. typos (GFAB, migroglia);

ii. qRT-PCR heatmap: it is not clear how the different markers vary along the time. The way it is presented looks like that some markers that are expected later in development (S100B) are rather expressed early, and downregulated later on;

iii. Specific cell marker (fold change plot): it is unclear what data is being plotted in this graph and what is the final conclusion related to this data.

3. There are multiple overstatements and generalizations throughout the text that need to be revised by the authors. See some examples below:

a) "Conventional cerebral organoid maturation approaches using spinners¹⁰ and shakers¹¹, promote self-neuronal organization and formation of cerebral ventricular spaces by providing agitation. However, these methods have major limitations, including altered nutrient diffusion and apoptotic zones due to lack of vascularization¹², immune cell deficiency, matrigel dependency, low reproducibility, scalability and high variability of the induced brain components and cells¹³." This statement should be rephrased as some of the limitations mentioned (e.g. apoptotic zones, lack of reproducibility) have already been addressed and overcome in prior studies (including the ones cited in this sentence).

b) The authors should be particularly careful about claiming "molecular and functional maturation of cerebral brain organoids". Not enough data is provided to make these claims and some of the available data is not of sufficient quality. For instance, colocalization of the postsynaptic marker, PSD95, and a presynaptic marker, demonstrated by confocal microscopy, would suggest that there are functional synapses present in the organoids, but would still be insufficient for concluding functional maturation of this organoids.

c) In the Discussion section there is a series of overstatements:

i. "(1) enriched cellular diversity (abundant neural/glial/ endothelial cell population)" > deeper characterization with higher quality images and more markers should be done to conclude this;

ii. "(2) structural brain morphogenesis (complex cortical layers, radia-glia organization, preplate splitting)" > no clear demonstration of brain morphogenesis; indeed, the lack of complex structure in brain organoids is one of the limitations of the field.

iii. "further functional neuronal identities (glutamatergic and hippocampal neurons) and synaptogenesis (postsynaptic interaction) during whole human brain development" > further analysis using more sophisticated technologies would be necessary to validate this claim.

iv. "GFAP+/S100B+ astrocytes and NEUN+/MAP2+ mature neurons were abundant on day 60 in RCCS organoids richest in cell diversity, both RCCS and μ -platform organoids expressed CD11b+/IBA1+ microglia and MBP+/OLIG2+ oligodendrocytes at high levels as of day 60." > further characterization with more markers and higher quality images are necessary to support this claim.

v. "RCCS organoids with enormous cell architectures, exhibited excellent cortical plate and SVZ architecture at day 120, where TUJ+ cells located at the basal side of the ventricle-like structures with TBR1+ cells located in the deep layer of cortical-like plates as well as well-defined progenitor zone

organization, neural identity and further neuronal differentiation with NESTIN+/TBR2+/PAX6+ and SOX1+/SOX2+ cells.” > it is not clear what the authors are referring to.

vi. “On the other hand, N-CAD+ apical epithelial and TTR+ choroid plexus epithelial cells were populated in RCCS organoids as of day 60, while both FOXG1+ forebrain, KROX20+ hindbrain and PROX1+ hippocampus related to midbrain and hindbrain identities were observed in μ -platform organoids at day 120.” > further characterization with more markers and higher quality images are necessary to support this claim.

Minor concerns:

1. The authors raise a valid point that hemodynamic forces are underexplored in the brain organoid field, despite the known biological relevance of mechanosensitive signals associated with cerebrospinal fluid flow during development. However, the authors do not explore this aspect in their data or discussion. It would be interesting to tie their data back to what is known about this in the neurodevelopmental field.

2. The authors hypothesize that microgravity and gravity-driven laminar flows enhance maturation of cerebral organoids recapitulating features of human embryonic cortical development. It would be interesting to discuss the findings that support this hypothesis.

3. “In terms of tissue structure, while the most dispersed organoid structure was observed in static culture and the most uniform distribution was observed in the shaker, the well-defined multi-patterned organoid morphologies with self-organization and desired cellular architectures in the inner-outer regions^{6,13,28} were observed in the RCCS microgravity bioreactor, followed by μ -platform and spinner systems (Fig. 2b,c,d,e,f).” It is not clear which parameters the authors used to say the latter. How was this quantified?

4. What does “neurons at different polarized states” mean?

5. Authors mentioned the presence of “cortical plate” in their organoids, but this is not clear from the images provided.

6. “Paired box protein 6 (PAX6) is a transcription factor expressed in neural stem cells and ventral forebrain radial glial progenitor cells. Although expressed in all brain regions, the highest expression level is reported at the cerebellum, which is involved in glial cell differentiation⁵⁵.” This sentence is confusing: it is unclear the relationship between glial cell differentiation and the cerebellum.

7. “CTIP2, SATB2, and TBR1 were upregulated at protein and RNA levels which indicated cortical neuronal differentiation in both RCCS and μ -platform organoids mainly on day 120 (Fig. 4, Supp. Fig. 3), supporting the spatial distribution and functional organization of cortical neurons⁶¹.” It has been demonstrated in 3D brain organoid cultures that generation of the different cortical neuron subtypes is not necessarily accompanied by structural/functional organization in the stereotypical layers of the human cerebral cortex (PMID: 24277810). Therefore, this sentence should be revised.

8. In the methods, the authors mentioned that some modifications were done to the organoid protocol, but they do not mention what was done.

Reviewer #2 (Remarks to the Author):

Saglam-Metiner et al. reported that RCCS microgravity bioreactor and μ -platform presented improvement of the harvestability, scalability, reproducibility, and survival of cerebral organoids, as well as functional maturation. It is assumed that comparing these various 3D culture methods with each other required a lot of effort, and the results should provide important information to the

scientific research field. However, their findings are just observations, and quantitative data are lacking. Statistical analysis of quantitative data is a basic requirement for scientific consideration. Furthermore, there are no data to allow the evaluation of physiological function, although the authors mentioned that their organoids have achieved functional maturity. There remain many issues to be resolved before reaching the conclusion the authors proposed.

Major points:

1. The authors should add the quantitative data to explain the improvement of harvestability, scalability, reproducibility, and survival of cerebral organoids to compare the respective culture conditions, and they should present the results of statistical analysis.
2. Although the authors described organoid-maturation, they only presented immunostaining and western blot data to show protein expression related to synapses such as PSD95. The authors should present the data of physiological analysis to show that their organoids have acquired functional maturation as neurons.
3. Although the authors mentioned that a layered structure of brain has formed, the figures showed no such structure but only a scattering of cells expressing markers for the cortical layers. The authors should correct the description.
4. The authors should provide the details of the equipment, and especially the μ -platform that was reported as being newly constructed in this paper.
5. The authors should compare the cell-type characteristics and maturation status in single-cell resolution methods like single-cell RNA sequence or detailed IFC investigation because organoid contains various types of brain cells. Quantification of these data will be helpful for understanding the beneficial points of the presented platform.
6. The authors attempted to characterize different culture platforms by hydrodynamic analysis. This analysis clarifies the characteristics of how the medium moves and can be considered to be useful data. However, what this analysis reveals is limited to phenomena outside of organoids, such as the distribution of shear stress. This analysis unfortunately does not help us to understand what physical effects occur inside individual organoids and, as a result, affect differentiation and maturation. Care should be taken so as not to overstate this point, and to improve the description to avoid misunderstandings.

Minor points

1. Signal/background ratios of images are not enough. Revise the setting when acquiring images.
2. The shape and marginal border of organoids are irregular, and these organoids appeared to be dying in Figures 3-6. The authors can investigate and describe what happens, e.g. cellular migration, cell death, unique differentiation, etc.
3. In Figure 5b, PROX1 appears to highlights the thalamus and cerebellum. Is that correct?
4. In Figure 6b, the band size of Ki-67 varies among different culture platforms.

Authors' Responses to the Review Comments

Journal: Communications Biology

Manuscript #: COMMSBIO-22-1668-T

Title of Paper: Spatio-temporal dynamics enhance cellular diversity, neuronal function and further maturation of human cerebral organoids

Thank you for giving the opportunity to submit our revised manuscript titled "Spatio-temporal dynamics enhance cellular diversity, neuronal function and further maturation of human cerebral organoids" to Communications Biology. We appreciate the editor and reviewers for their valuable time and effort in our manuscript. We have revised our paper based on the insightful comments of the reviewers and highlighted the changes in the manuscript. We hope that the revised manuscript can meet the journal's publication requirements.

Reviewer #1 (Remarks to the Author):

In this manuscript, Saglam-Metiner et al. describe two novel engineering approaches (RCCS and u-platform) with the goal to improve harvestability, scalability, reproducibility and survival of human brain organoids during long term cultures. The authors start by elegantly demonstrating that the proposed models (RCCS and u-platform) exhibit less shear stress when compared to traditional cultures on shakers or spinner flasks. Moreover, the shear stress distribution in RCCS and u-platform is a lot more homogeneous than when compared to the current available systems (shakers and spinners). These novel methods offer a great potential to further improve quality and durability of human brain organoids in vitro, which have been negatively impacted by the limitations associated with current culture conditions. Despite the great potential, the manuscript needs significant improvement on the characterization of differentiation and maturation of cells within the human brain organoids to conclusively assess the value of these approaches to advance the field.

Thanks for the insightful comments, which helped to increase the quality of the manuscript. Please find our point-by-point responses to each of the comments raised.

Major concerns:

1. Cell composition variability is known to be a limitation of some human brain organoid protocols. In this study, the authors refer to reproducibility in terms of organoid size, which does not infer much about the quality of the differentiation and cell composition. Therefore, conclusions about reproducibility should take other parameters in consideration, such as cellular composition and cellular organization, instead of just organoid size

We appreciate the reviewer's suggestion and agree that there should be additional data to strengthen the conclusions of the study. Therefore, we performed fluorescence-activated cell sorting (FACS)

analysis of microglia and astrocytes from the RCCS organoids to support cellular composition in a revised text as Fig. 3d. Method details is given as follows;

Initially, for single-cell suspension from RCCS organoids, three organoids were freshly harvested from vessels, collected to tube, washed with 1x PBS and immersed in 1 mL DMEM/F12 included papain (18.6 U/mL, P3125, Sigma-Aldrich) and DNase 1 (337 U/mL, EN0521, Thermo Fisher) at 37°C for 30 min on shaker. It was mechanically pipetted and vortexed at ten-minute intervals under sterile conditions. Next, 2% FBS was added to stop the enzymatic reaction and the resulting single-cell suspension was centrifuged at 400 rfc for 5 min. An additional incubation was made in PBS buffer (included 2 mM EDTA, 1% FBS and 337 U/mL DNase 1, pH 7.4) at RT for 15 min, and single-cells were passed through a 70 µm cell-strainer. Then, single-cells were labeled with CD45 and CD11b primary antibodies for microglia sorting and were labeled with GFAP primary antibody for astrocyte cell sorting (Supp. Table 4) in FACS buffer (filtered 1xPBS included 2%BSA, 50 mM EDTA, and 5 ng/mL SCF, pH 7.4) at RT for 1 h. Finally, after centrifugation and 1xPBS washing, single-cells were labeled with Alexa fluor-conjugated secondary antibodies (Supp. Table 4) in the FACS buffer and incubated for 1 h at RT. To detect the dead cells, DAPI-staining was performed after 1xPBS washing and cells were sorted/gated that were alive, single, and CD11b+/CD45+ or GFAP+ by the FACSAria III with 100 micron nozzle tip^{42,43,118}.

Additionally, we added a graph to highlight the harvestability results in Fig. 2b and detailed all of the results in the text under the heading “Laminar flow increases reliable production of high-quality organoids with the highest harvestability, reproducibility and reduced batch-to-batch variabilities” as follows;

On the 15th day of culture, generated organoids (derived from two different passage numbers of iPSCs at two different times) were transferred to RCCS, µ-platform, spinner and shaker, along with static culture as the control group (considered as day 0 of maturation), in order to determine the best dynamic system for physical and functional maturation of cerebral organoids, which were cultured for 120 days. Organoid size distribution variability is known to be a limitation of traditional human brain organoid protocols, as well as cellular composition and organization. Therefore, organoid reproducibility and batch-to-batch variability was firstly examined in terms of harvestability, macroscopic-microscopic observations and organoid size distribution (Fig. 2b,c,d,e,f,g). As such, cellular composition and cellular organization were assessed with cell sorting, immunostaining, western blot, qRT-PCR, TUNEL and glutamate secretion analyses for organoids sampled on various days (30, 60 and 120th days).

2. Overall presentation and description of data should be significantly improved. There is insufficient characterization that limits interpretation and conclusions. Moreover, proportion of cell types and protein expression patterns present in the study are inconsistent with what is known in the field. This data should be revised and validated more carefully. Below are some examples that require careful revision:

a) Brightfield images (Fig. 2) and immunostainings throughout the manuscript raise concerns about the organoid differentiation quality. It is very hard to assess the quality of differentiation by these images and it is not very clear what information the authors want to convey with the H&E staining (Fig. 2). Additionally, the transparent regions observed in organoids from most conditions resemble cystic formation associated with off-target differentiation. Therefore, the authors should demonstrate that the

organoids are indeed cortical organoids. This can be shown by high-quality images demonstrating a high proportion of FOXG1+ cells. FOXG1 is a forebrain marker that is expressed early during cortical development (PMID: 18983967, PMID: 24277810, PMID: 23995685). Moreover, the authors should include high-quality images including other standard markers to confirm successful cortical differentiation (e.g. EMX1, PAX6, SOX2, TBR1, CTIP2, SATB2), as well as verify the presence of rosette and neural tube-like structures in multiple timepoints

The quality of all figures has been increased and revised in the illustration program. More clear information about H&E staining (images with small magnification have been removed to avoid misunderstanding) and transparent regions added to text with white arrows to indicate organized cell nucleus and neural rosette-like structures in Fig. 2c,d,e,f,g. As well as N-CAD/FOXG1 image replacement for RCCS organoids in Fig.4a, a new staining was carried out and high-quality images demonstrating high proportion of FOXG1+ and PAX6+ cells were added to assess the quality of differentiation in Fig.6a. Also, white arrows added to indicate TBR1+/PAX6+/SOX2+/FOXG1+ and DAPI+ neural rosette and neural tube-like structures in Fig. 4a, 6a,b and Supp. Fig. 2. Next, a schematic illustration and new literature information was added to the text to show cellular organization of matured cerebral organoids in MZ, CP, oSVZ, SVZ, VZ zones in Fig.4b. Additionally, yellow dashed lines were added to indicate cortical plate layer borders in Fig. 4a,5a, between the early-born neurons expressed CTIP2 in the deep-layer and the late-born neurons expressed SATB2 in the superficial upper-layer.

b) The authors claim that both astrocytes (GFAP+) and neurons (NEUN+) are present since day 60 of organoid development. Previous studies with human brain organoids have demonstrated that differentiation in these in vitro 3D models recapitulate endogenous development where gliogenesis happens after neurogenesis. In one of these papers, astroglia are not seen before 6 months of development in vitro (PMID: 28445462). It is possible that modifications done by the authors to the protocol change this pattern, but it is recommended that higher magnification images, including more markers per cell type, are included to better illustrate the presence of these cells in their models.

Thanks for pointing this out. The quality of all figures has been increased and revised in the illustration program. The GFAP+/S100B+ astrocytes and NEUN+/MAP2+ mature neurons were examined in detail by immunofluorescence, western blot or qRT-PCR analysis. Additionally, we performed FACS analysis of GFAP+ astrocytes from the RCCS organoids to support the presence of more cellular composition and added the results to revised text as Fig. 3d with $1.3\% \pm 0.4$ GFAP+ cell sorting rate.

c) The authors claim presence of cell types such as microglia and oligodendrocytes that are usually not present in most protocols. It is also recommended to include higher magnification images of these cells with more markers (microglia: IBA1, P2RY12, TMEM119; oligodendrocytes: MBP, OLIG2, PLP) in order to validate their presence in the organoids.

We added the following paragraph to the text;

To further interrogate the presence of microglial cells, the ratio of CD45/CD11b double positive cells in RCCS organoids was found to be $1.3\% \pm 0.4$ with FACS analysis, indicating the positive effect of microgravity on mesodermal-microglial cell differentiation (Fig. 3d). In agreement with our findings, Ormel et al. reported that CD11b+ microglia constituted $5\% \pm 2.8$ of the entire single-cell suspension with MACS analysis, as well as CD11b+/CD45+ microglia population was $0.83\% \pm 0.3$ with FACS for 38-52 days old cerebral organoid⁴². Shiraki et al. showed that the ratio of

*CD11b/CD45 double positive PAX6-positive microglia cells was 0.33%±0.12 in 4 week differentiation of hiPSC-derived ocular organoids*⁴³.

d) Staining for the postsynaptic marker, PSD95, seems unspecific in the representative images and would require co-staining with a presynaptic marker (GABA-A) and imaging using a confocal microscope.

Thanks for the suggestion. Besides, postsynaptic PSD95 and presynaptic vGLUT1 staining, we additionally performed immunofluorescence staining with a presynaptic marker GABA-A to indicate GABAergic interneurons in organoids, and obtained high-quality images under the confocal microscope. GABA-A immunofluorescence staining is given in the results section in Fig. 5a. and details about GABA-A is provided in the results section as follows. Moreover, western blot analysis was performed to support results in Fig. 5c.

During early corticogenesis, bursts of action potentials cause spreading of giant waves of calcium influxes through the developing cortex, which is described as giant depolarizing potentials (GDPs), depending both on excitatory glutamate and gamma-aminobutyric acid (GABA) inputs. Glutamate-dependent GDP-like events were reported in neuronal organoids. Reduced GDPs in >40 day organoids with GABA polarity switch were regarded as indicators of progressive neuronal network maturation⁷⁸. On the other hand, the glutamate level that secreted from microglia was measured around 20 μ M in healthy in vitro neuronal cultures^{79,80}. Given that, the change in the uptake and consequently release rate of extracellular glutamate in 60 day RCCS and μ -platform organoids and growth in organoid sizes might be associated with more mature states. One of the distinguishing features of neuronal circuit and maturation is the switch of excitatory to inhibitory GABAergic neurotransmission. GABA-A, a presynaptic GABAergic interneuron receptor that modulates neurotransmitter release in both peripheral and central synapses^{7,78,81-83}, was more expressed in both RCCS and μ -platform organoids with CTIP2+ deep-layer of cortical neurons as of day 60 (Fig. 5a,b,c). Thus, the neuronal network complexity was validated in RCCS and μ -platform organoids with the presence of both presynaptic glutamatergic VGLUT1+ and GABAergic GABA-A+ neurons which are critically involved in neuronal network oscillations, as well as postsynaptic PSD95+ neurons, GFAP+/S100B+ astrocytes and MBP+/OLIG2+ oligodendrocytes.

e) KI67 staining should be nuclear; however in Fig. 6a the staining seems to be outside of the nucleus. Moreover, specifically in this panel, representative organoids depicted in the RCCS and μ -platform raise concern about the quality of differentiation, as broad morphology seems cystic and irregular.

Thanks for pointing this out, we agree that the Ki-67 staining should be nuclear. Therefore, we repeated Ki-67 immunofluorescence staining for all groups and obtained images that resulted in more nuclear staining than the previous stainings. We revised Ki-67 images in Fig. 6a and the quality of all figures has been increased and revised in the illustration program. Additionally, to support the immunofluorescence staining results, western blot and qRT-PCR analysis were performed for the Ki-67 marker, which can be found in Fig. 6b,c and added to the text.

f) Font size in Fig. 1 should be increased in order to be seen.

The font size in Fig.1 was increased in order to be seen

g) Other examples within Fig. 3

i. typos (GFAB, migroglia)

Thanks, we corrected all typos throughout the text

ii. qRT-PCR heatmap: it is not clear how the different markers vary along the time. The way it is presented looks like that some markers that are expected later in development (S100B) are rather expressed early, and downregulated later on

qRT-PCR heatmap illustrations are revised as a system-based grouping to explicit their changes over time. The heatmaps were also colored with a monochromatic gradient that increases in intensity from low to high expression to illustrate the expression changes more clearly.

iii. Specific cell marker (fold change plot): it is unclear what data is being plotted in this graph and what is the final conclusion related to this data.

Thanks, mentioned graph is removed to avoid misunderstanding.

3. There are multiple overstatements and generalizations throughout the text that need to be revised by the authors. See some examples below:

a) “Conventional cerebral organoid maturation approaches using spinners¹⁰ and shakers¹¹, promote self-neuronal organization and formation of cerebral ventricular spaces by providing agitation. However, these methods have major limitations, including altered nutrient diffusion and apoptotic zones due to lack of vascularization¹², immune cell deficiency, matrigel dependency, low reproducibility, scalability and high variability of the induced brain components and cells¹³.” This statement should be rephrased as some of the limitations mentioned (e.g. apoptotic zones, lack of reproducibility) have already been addressed and overcome in prior studies (including the ones cited in this sentence).

Thank you for suggestion, the sentence is rephrased as follows;

However, these methods have major limitations, including altered nutrient diffusion and apoptotic zones due to lack of vascularization, immune cell deficiency, matrigel dependency, low reproducibility, scalability and high variability of the induced brain components and cells¹². Different strategies have been developed to prevent these limitations such as by co-culturing with human umbilical vascular endothelial cells (HUVECs)¹³ and human iPSC-derived endothelial cells not only to enhance oxygen/nutrient diffusion but also to leverage neural differentiation, migration and circuit formation during development¹⁴.

b) The authors should be particularly careful about claiming “molecular and functional maturation of cerebral brain organoids”. Not enough data is provided to make these claims and some of the available data is not of sufficient quality. For instance, colocalization of the postsynaptic marker, PSD95, and a presynaptic marker, demonstrated by confocal microscopy, would suggest that there are functional synapses present in the organoids, but would still be insufficient for concluding functional maturation of this organoids

We appreciate the reviewer’s insightful suggestion and agree with this. Thus, we performed additional glutamate analysis to examine molecular and functional maturation of RCCS and platform organoids as depicted in Fig. 5d and details are given in the methods and results sections as follows..

To ascertain advanced maturation of RCCS and μ -platform organoids, we examined the release of glutamate. For this, samples of the medium (one week of used) were collected on day 60 of maturation process and analyzed with a colorimetric glutamate assay kit (MAK330-1K, Sigma-Aldrich), according to the manufacturer's instructions. Samples of fresh cerebral organoid differentiation medium were also run as negative controls.

VGLUTs are responsible for the vesicular accumulation of glutamate, the major excitatory neurotransmitter that plays critical roles in neuronal signaling and cortical development, which is uploaded into synaptic vesicles within presynaptic terminals before undergoing regulated release at the synaptic cleft^{76,77}. Therefore, we decided to examine the cumulative glutamate levels in one week of used maturation medium of RCCS and μ -platform organoids at days 30 and 60 to evaluate further maturation. Significant time-dependent increases were noted in glutamate levels for both groups ($p < 0.0001$ and $p < 0.01$, respectively), even more in RCCS organoids as $165 \pm 2.8 \mu\text{M}$ ($p < 0.001$) (Fig. 5d). During early corticogenesis, bursts of action potentials cause spreading of giant waves of calcium influxes through the developing cortex, which is described as giant depolarizing potentials (GDPs), depending both on excitatory glutamate and gamma-aminobutyric acid (GABA) inputs. Glutamate-dependent GDP-like events were reported in neuronal organoids. Reduced GDPs in >40 day organoids with GABA polarity switch were regarded as indicators of progressive neuronal network maturation⁷⁸. On the other hand, the glutamate level that secreted from microglia was measured around $20 \mu\text{M}$ in healthy in vitro neuronal cultures^{79,80}. Given that, the change in the uptake and consequently release rate of extracellular glutamate in 60 day RCCS and μ -platform organoids and growth in organoid sizes might be associated with more mature states.

Besides the postsynaptic PSD95 and presynaptic vGLUT1 staining, we newly performed immunofluorescence staining with a presynaptic marker GABA-A to indicate GABAergic interneurons in organoids, and obtained high-quality images under the confocal microscope, which are depicted in Fig. 5a. Additionally, western blot analysis was performed to support result, which can be seen in Fig. 5c.

c) In the Discussion section there is a series of overstatements:

i. "(1) enriched cellular diversity (abundant neural/glia/ endothelial cell population)" > deeper characterization with higher quality images and more markers should be done to conclude this;

Thanks for your comments. Besides rephrasing the sentences in the discussion section, more in-depth characterization was provided in the revised text with high-quality images and FACS, IF, WB and qRT-PCR analyses, as mentioned above.

ii. "(2) structural brain morphogenesis (complex cortical layers, radia-glia organization, preplate splitting)" > no clear demonstration of brain morphogenesis; indeed, the lack of complex structure in brain organoids is one of the limitations of the field.

A clear demonstration was provided with schematic illustration of cellular organization of matured cerebral organoids in Fig. 4b. Additionally, high-quality confocal images, neural rosette tube-like structures indicated by white arrows and CP layer borders indicated by yellow dashed lines were added to the text.

iii. “further functional neuronal identities (glutamatergic and hippocampal neurons) and synaptogenesis (postsynaptic interaction) during whole human brain development” > further analysis using more sophisticated technologies would be necessary to validate this claim.

For further analysis, glutamate measurements, additional IF and WB analysis for GABA-A marker were carried out to validate functional maturation of organoids.

iv. “GFAP+/S100B+ astrocytes and NEUN+/MAP2+ mature neurons were abundant on day 60 in RCCS organoids richest in cell diversity, both RCCS and μ -platform organoids expressed CD11b+/IBA1+ microglia and MBP+/OLIG2+ oligodendrocytes at high levels as of day 60.” > further characterization with more markers and higher quality images are necessary to support this claim.

In addition to IF, WB and qRT-PCR analysis, we performed further characterization of cells with FACS analysis. New results have been added to the relevant sections in the revised text.

v. “RCCS organoids with enormous cell architectures, exhibited excellent cortical plate and SVZ architecture at day 120, where TUJ+ cells located at the basal side of the ventricle-like structures with TBR1+ cells located in the deep layer of cortical-like plates as well as well-defined progenitor zone organization, neural identity and further neuronal differentiation with NESTIN+/TBR2+/PAX6+ and SOX1+/SOX2+ cells.” > it is not clear what the authors are referring to.

The sentences are rephrased in the discussion section, clear demonstration was provided with schematic illustration of cellular organization of matured cerebral organoids in Fig. 4b, as mentioned above.

vi. “On the other hand, N-CAD+ apical epithelial and TTR+ choroid plexus epithelial cells were populated in RCCS organoids as of day 60, while both FOXG1+ forebrain, KROX20+ hindbrain and PROX1+ hippocampus related to midbrain and hindbrain identities were observed in μ -platform organoids at day 120.” > further characterization with more markers and higher quality images are necessary to support this claim.

Thank you, we performed further characterization with higher quality revised images and more markers (β -catenin) and new IF, WB and qRT-PCR analysis (for FOXG1). Furthermore, new β -catenin marker results have been added to the revised text as follows;

Vascularization in cerebral organoids was reported to up-regulate the Wnt/ β -catenin signaling and increase Ki-67 cell proliferation marker. Endothelial-like cells in vascularized organoids signal to neural stem cells, regulate their self-renewal and differentiation into neurons during CNS development⁸⁷. Additionally, the adherens junctions control vRG cell's self-renewal, proliferation, differentiation and survival via active Wnt/ β -catenin/N-cadherin signaling⁸⁸. We also found that both protein and RNA expression levels of β -catenin were more elevated in microgravity driven RCCS organoids and low shear stress induced μ -platform organoids on day 120 (Fig 6a,b,c, Supp. Table 1) as in N-CAD levels, while neural/glial cell diversity and CP-like structures were observed (white arrows indicate PAX6+ neural rosette and neural tube-like structures).

Minor concerns:

1. The authors raise a valid point that hemodynamic forces are underexplored in the brain organoid field, despite the known biological relevance of mechanosensitive signals associated with cerebrospinal fluid flow during development. However, the authors do not explore this aspect in their data or discussion. It would be interesting to tie their data back to what is known about this in the neurodevelopmental field.

Within the reviewer's suggestion, we have detailed the results and discussion parts to be more explanatory. Minor concerns 1 and 2 are considered together and explained below

2. The authors hypothesize that microgravity and gravity-driven laminar flows enhance maturation of cerebral organoids recapitulating features of human embryonic cortical development. It would be interesting to discuss the findings that support this hypothesis.

Recently, the transcriptomic analyses of cerebral organoids showed that mechanotransduction-associated genes including integrins, β -catenin, Wnt, and Delta-like pathway genes change under physical stress conditions⁹⁸. Thus, organoids exposed to varying magnitudes of hydrodynamic forces are expected to display differences in structural and functional features during development. As such, cell proliferation of hPSCs has been shown to increase under simulated microgravity culture conditions with significantly higher expression levels of Ki-67 compared to 1g culture condition, leading to enhanced self-renewal, as revealed by the increased protein levels of the core set of pluripotent transcription factors⁹⁹. In another study, human embryonic stem cell (hESC)-derived forebrain-specific neural-cortical organoids were shown to have the highest expression of Ki-67 when cultured under RCCS conditions after day 14 of the generation process compared to organoids cultured in static and other days of the generation in RCCS¹⁰⁰. These results are in agreement with our study and support the increased expression levels of Ki-67 in RCCS organoids. Also, the use of RCCS under microgravity conditions in embryonic body formation from ESCs has been shown to promote more homogeneous EB formation and endoderm differentiation by modulating the Wnt/ β -catenin pathway¹⁰¹. On the other hand, the most severe subtype of spina bifida, which is one of the neural tube defects that starts in the 4th week of pregnancy, occurs when the spinal cord is exposed to shear stress from amniotic fluid^{102,103}. This negative effect of shear stress on brain and spinal cord formation during embryonic development highlights the importance of shear stress during organoid maturation.

3. “In terms of tissue structure, while the most dispersed organoid structure was observed in static culture and the most uniform distribution was observed in the shaker, the well-defined multi-patterned organoid morphologies with self-organization and desired cellular architectures in the inner-outer regions^{6,13,28} were observed in the RCCS microgravity bioreactor, followed by μ -platform and spinner systems (Fig. 2b,c,d,e,f).” It is not clear which parameters the authors used to say the latter. How was this quantified?

More clear information about H&E staining added to text as follows and white arrows are added to Fig.2c,d,e,f,g. to allow better understanding (images with small magnification have been removed to avoid complexity)

In terms of tissue structure, while the most dispersed organoid structure was observed in static culture and the most uniform cell distribution (unorganized) was observed in the shaker, the well-defined multi-patterned organoid morphologies with self-organization and desired cellular

architectures in the inner-outer regions^{6,12,28} were more observed in the RCCS microgravity bioreactor (sequential organized cell nucleus and neural rosette-like structure indicated by white arrows), followed by μ -platform and spinner systems (Fig. 2c,d,e,f,g).

Additionally, quantification was made with a graph to highlight the harvestability results in Fig.2b and organoid size graphs in Fig.2c,d,e,f,g.

4. What does “neurons at different polarized states” mean?

This statement is detailed as follows;

During human embryonic brain development, neurons and glial cells are derived from neural stem/progenitor cells that generate specific brain regions and cortical layers. The derivation is achieved by an asymmetric spatial organization of cells and different components, explained as cell polarity. In the polarization process, the ventricular zone (VZ) formed by neuroepithelial cells is followed by the subventricular zone (SVZ), where apical progenitors form basal progenitors, and then cortical plate (CP) by apical and basal progenitors forming neurons⁵⁰. The degree of efficient differentiation is determined by the expression of neurons at different polarized states and progenitor specific markers.

5. Authors mentioned the presence of “cortical plate” in their organoids, but this is not clear from the images provided.

As well as replacement of N-CAD/FOXP1 images with better resolutions for RCCS organoids in Fig.4a, a new staining was carried out and high-quality images demonstrating a high proportion of FOXP1+ and PAX6+ cells were added to assess the quality of neuronal cell differentiation in Fig.6a. Also, white arrows added to indicate TBR1+/PAX6+/SOX2+/FOXP1+ and DAPI+ neural rosette and neural tube-like structures in Fig. 4a, 6a,b and Supp. Fig. 2. Next, a schematic illustration and new literature information was added to text to show cellular organization of matured cerebral organoids as Fig.4b. Then, yellow dashed lines were added to indicate cortical plate layer borders in Fig.4a,5a, between the early-born neurons expressed CTIP2 in the deep-layer and the late-born neurons expressed SATB2 in the superficial upper-layer.

6. “Paired box protein 6 (PAX6) is a transcription factor expressed in neural stem cells and ventral forebrain radial glial progenitor cells. Although expressed in all brain regions, the highest expression level is reported at the cerebellum, which is involved in glial cell differentiation⁵⁵.” This sentence is confusing: it is unclear the relationship between glial cell differentiation and the cerebellum.

The text is revised as follows;

Paired box protein 6 (PAX6) is a transcription factor expressed in neural stem cells along with ventral forebrain radial glial progenitor cells and has an important role in the differentiation of radial glial cells located in different regions of the CNS, such as the cerebral cortex, cerebellum, forebrain and hindbrain⁵⁹.

7. “CTIP2, SATB2, and TBR1 were upregulated at protein and RNA levels which indicated cortical neuronal differentiation in both RCCS and μ -platform organoids mainly on day 120 (Fig. 4, Supp. Fig. 3), supporting the spatial distribution and functional organization of cortical neurons⁶¹.” It has been demonstrated in 3D brain organoid cultures that generation of the different cortical neuron subtypes is not necessarily

accompanied by structural/functional organization in the stereotypical layers of the human cerebral cortex (PMID: 24277810). Therefore, this sentence should be revised.

The sentence is revised as follows:

CTIP2, SATB2, and TBR1 were upregulated at protein and RNA expression levels in both RCCS and μ -platform organoids mainly on day 120 (Fig. 4, Supp. Fig. 3, Supp. Table 5), showing an inside-out pattern of CP (yellow dashed lines indicate CP layer borders). Early-born neurons expressed CTIP2 and TBR1 in the deep-layer, whereas late-born neurons expressed SATB2 in the superficial upper-layer with the obvious spatial zone separation^{65,66}

8. In the methods, the authors mentioned that some modifications were done to the organoid protocol, but they do not mention what was done.

The sentence is revised as follows to avoid any confusion;

The cerebral organoid generation steps were based on the detailed method of Lancaster and Knoblich¹¹⁶ with some minor modifications regarding the durations in consecutive steps starting from maturation, in which dynamic systems and flow rates are different from the applied method.

Reviewer #2 (Remarks to the Author):

Saglam-Metiner et al. reported that RCCS microgravity bioreactor and μ -platform presented improvement of the harvestability, scalability, reproducibility, and survival of cerebral organoids, as well as functional maturation. It is assumed that comparing these various 3D culture methods with each other required a lot of effort, and the results should provide important information to the scientific research field. However, their findings are just observations, and quantitative data are lacking. Statistical analysis of quantitative data is a basic requirement for scientific consideration. Furthermore, there are no data to allow the evaluation of physiological function, although the authors mentioned that their organoids have achieved functional maturity. There remain many issues to be resolved before reaching the conclusion the authors proposed.

Thanks for the comments and suggestions, which helped to increase the quality of the manuscript. Please find our point-by-point responses to each of the comments raised.

Major points:

1. The authors should add the quantitative data to explain the improvement of harvestability, scalability, reproducibility, and survival of cerebral organoids to compare the respective culture conditions, and they should present the results of statistical analysis.

Thank you for the comment. We added a graph and statistical analysis to highlight the harvestability results in Fig. 2b. The sentences are revised in the text as follows;

On the 15th day of culture, generated organoids (derived from two different passage numbers of iPSCs at two different times) were transferred to RCCS, μ -platform, spinner and shaker, along with static culture as the control group (considered as day 0 of maturation), in order to determine the

best dynamic system for physical and functional maturation of cerebral organoids, which were cultured for 120 days. Organoid size distribution variability is known to be a limitation of traditional human brain organoid protocols, as well as cellular composition and organization. Therefore, organoid reproducibility and batch-to-batch variability was firstly examined in terms of harvestability, macroscopic-microscopic observations and organoid size distribution (Fig. 2b,c,d,e,f,g). As such, cellular composition and cellular organization were assessed with cell sorting, immunostaining, western blot, qRT-PCR, TUNEL and glutamate secretion analyses for organoids sampled on various days (30, 60 and 120th days).

We used the ImageJ programme to quantify apoptotic zone vs nucleus, data were then calculated with GraphPad Prism 8.3.0. The quantitative results of apoptotic zone % in Fig. 6d with statistical analysis were added to the revised text.

With respect to the scalability, the sentences in the discussion section were corrected as follows to eliminate the ambiguity;

Although scale-up was not planned for this study, we presume that about 1000 matured organoids can be harvested in one batch, if we scale up to 500 ml RCCS STLVs, as such the μ -platform can be multiplied by connecting in a parallel or serial manner for scale-up purposes.

2. Although the authors described organoid-maturation, they only presented immunostaining and western blot data to show protein expression related to synapses such as PSD95. The authors should present the data of physiological analysis to show that their organoids have acquired functional maturation as neurons.

Thanks for the suggestion. Besides, postsynaptic PSD95 and presynaptic vGLUT1 staining, we additionally performed immunofluorescence staining with a presynaptic marker GABA-A to indicate GABAergic interneurons in organoids, and obtained high-quality images under the confocal microscope. GABA-A immunofluorescence staining is given in the results section in Fig. 5a. and details about GABA-A is provided in the results section as follows. Moreover, western blot analysis was performed to support results in Fig. 5c.

During early corticogenesis, bursts of action potentials cause spreading of giant waves of calcium influxes through the developing cortex, which is described as giant depolarizing potentials (GDPs), depending both on excitatory glutamate and gamma-aminobutyric acid (GABA) inputs. Glutamate-dependent GDP-like events were reported in neuronal organoids. Reduced GDPs in >40 day organoids with GABA polarity switch were regarded as indicators of progressive neuronal network maturation⁷⁸. On the other hand, the glutamate level that secreted from microglia was measured around 20 μ M in healthy in vitro neuronal cultures^{79,80}. Given that, the change in the uptake and consequently release rate of extracellular glutamate in 60 day RCCS and μ -platform organoids and growth in organoid sizes might be associated with more mature states. One of the distinguishing features of neuronal circuit and maturation is the switch of excitatory to inhibitory GABAergic neurotransmission. GABA-A, a presynaptic GABAergic interneuron receptor that modulates neurotransmitter release in both peripheral and central synapses^{7,78,81-83}, was more expressed in both RCCS and μ -platform organoids with CTIP2+ deep-layer of cortical neurons as of day 60 (Fig. 5a,b,c). Thus, the neuronal network complexity was validated in RCCS and μ -platform organoids with the presence of both presynaptic glutamatergic VGLUT1+ and GABAergic GABA-

A+ neurons which are critically involved in neuronal network oscillations, as well as postsynaptic PSD95+ neurons, GFAP+/S100B+ astrocytes and MBP+/OLIG2+ oligodendrocytes.

Additionally, we performed glutamate analysis to examine molecular and functional maturation of RCCS and platform organoids as depicted in Fig.5d and details are given in the methods and results section, as mentioned before.

3. Although the authors mentioned that a layered structure of brain has formed, the figures showed no such structure but only a scattering of cells expressing markers for the cortical layers. The authors should correct the description.

Within the reviewer's valuable suggestion, a new staining of FOXG1+ and PAX6+ cells were added to assess the quality of neuronal differentiation in Fig.6a and white arrows added to indicate TBR1+/PAX6+/SOX2+/FOXG1+ and DAPI+ neural rosette and neural tube-like structures in Fig. 4a,6a,b and Supp. Fig. 2. Also, a schematic illustration and new literature information was added to text to show cellular organization of matured cerebral organoids with respect to MZ, CP, oSVZ, SVZ, VZ zones as depicted in Fig.4b. Yellow dashed lines were added to indicate cortical plate layer borders in Fig.4a,5a, between the early-born neurons expressed CTIP2 in the deep-layer and the late-born neurons expressed SATB2 in the superficial upper-layer.

4. The authors should provide the details of the equipment, and especially the μ -platform that was reported as being newly constructed in this paper.

Thanks, "Supp. Fig. 1" has been revised by adding all system illustrations and detailed with the size/volume information used in the simulation.

5. The authors should compare the cell-type characteristics and maturation status in single-cell resolution methods like single-cell RNA sequence or detailed IFC investigation because organoid contains various types of brain cells. Quantification of these data will be helpful for understanding the beneficial points of the presented platform.

We appreciate the reviewer's insightful suggestion. Besides detailed IF, WB and qRT-PCR analysis, the manuscript has been much improved within the newly performed quantitative FACS analysis to examine important types of brain cells such as CD45+/CD11b+ microglia and GFAP+ astrocyte in RCCS organoids, which is presented in Fig. 3d

6. The authors attempted to characterize different culture platforms by hydrodynamic analysis. This analysis clarifies the characteristics of how the medium moves and can be considered to be useful data. However, what this analysis reveals is limited to phenomena outside of organoids, such as the distribution of shear stress. This analysis unfortunately does not help us to understand what physical effects occur inside individual organoids and, as a result, affect differentiation and maturation. Care should be taken so as not to overstate this point, and to improve the description to avoid misunderstandings.

Thanks for pointing this out, we agree that the flow simulation just characterizes the hydrodynamics (velocity, shear stress, flow distribution, etc.) acting on the organoids. However, the purpose of this work was not related to clarifying the underlying mechanisms of mechanotransduction signaling. Herein, the effects of different external fluid dynamics on maturation in cerebral organoids were

examined with gene and protein expression levels. Furthermore, we believe that this work will shed light on future studies that will show which pathways responsible for mechanotransduction are activated by transcriptomic analyzes. With your comment, we avoided overstatements in the text and added more in-depth discussion as follows;

Physical forces, such as shear stress, gravity and cyclic stretch affect mechanosensitive pathways or change the expression levels²². Recently, the transcriptomic analyses of cerebral organoids showed that mechanotransduction-associated genes including integrins, β -catenin, Wnt, and Delta-like pathway genes change under physical stress conditions⁹⁸. Thus, organoids exposed to varying magnitudes of hydrodynamic forces are expected to display differences in structural and functional features during development. As such, cell proliferation of hPSCs has been shown to increase under simulated microgravity culture conditions with significantly higher expression levels of Ki-67 compared to 1g culture condition, leading to enhanced self-renewal, as revealed by the increased protein levels of the core set of pluripotent transcription factors⁹⁹. In another study, human embryonic stem cell (hESC)-derived forebrain-specific neural-cortical organoids were shown to have the highest expression of Ki-67 when cultured under RCCS conditions after day 14 of the generation process compared to organoids cultured in static and other days of the generation in RCCS31. These results are in agreement with our study and support the increased expression levels of Ki-67 in RCCS organoids. Also, the use of RCCS under microgravity conditions in embryonic body formation from ESCs has been shown to promote more homogeneous EB formation and endoderm differentiation by modulating the Wnt/ β -catenin pathway¹⁰⁰. On the other hand, the most severe subtype of spina bifida, which is one of the neural tube defects that starts in the 4th week of pregnancy, occurs when the spinal cord is exposed to shear stress from amniotic fluid^{101,102}. This negative effect of shear stress on brain and spinal cord formation during embryonic development highlights the importance of shear stress during organoid maturation.

Minor points

1. Signal/background ratios of images are not enough. Revise the setting when acquiring images

Thanks for the comment. The quality and resolution of all figures has been increased and revised in the illustration program. If required, all figures can be uploaded in tiff format, separately from the manuscript.

2. The shape and marginal border of organoids are irregular, and these organoids appeared to be dying in Figures 3-6. The authors can investigate and describe what happens, e.g. cellular migration, cell death, unique differentiation, etc.

Thanks for the comment. For better visualization of organoid sections, the quality and resolution of all figures has been increased and revised in the illustration program. Also, cryo-sections of organoids were obtained at 10-20 μ m thicknesses and all confocal images were 10x or 25x magnification.

3. In Figure 5b, PROX1 appears to highlights the thalamus and cerebellum. Is that correct?

Thanks for the comments. Lavado and Oliver (2006), mentioned that; "At an early postnatal stage, PROX1 expression is mainly detected in several nuclei of the thalamus, the cerebellum, and the hippocampus. In adulthood, PROX1 expression remains only in the hippocampus and cerebellum." So, the sentence is revised with this information in text.

4. In Figure 6b, the band size of Ki-67 varies among different culture platforms

Within the reviewer's suggestion, Western blot analysis for Ki-67 was repeated and Fig.6b in the results section was revised as a result of repeated Western blot analysis. The Ki-67 protein band can be seen at around 300 kDa (based on the supplier and the dilution ratio of primary antibody used).

REVIEWERS' COMMENTS:

Reviewer #2 (Remarks to the Author):

The authors responded this reviewer's comments appropriately.

Final Revision Instructions

*To the Author— Please review the editorial comments and requests below and confirm that changes have been made in the manuscript in the right-hand column. **This document must be uploaded** as a related manuscript file.*

Please see our final file submission checklist for information about submitting your revised documents.

Files and General Policies	
Main manuscript file must be in Microsoft Word or LaTeX format. LaTeX and Tex article source files must be accompanied by the compiled PDF for reference. The bibliography must be submitted separately (as a .bib file) or contained within the .tex file.	
Each Figure must be provided as a separate file and must be supplied whole, with all panels included in a single document. Figures should be provided at a minimum resolution of 300 dpi at final size. Figure files must only contain images (please also leave out labels such as “Figure 1” etc). Figure captions must instead be included within the main manuscript file, grouped together at the end of the document.	
All figures, tables, and supplementary items must be cited in the manuscript and numbered in the order in which they appear.	
Please ensure that all equations are supplied in an editable format upon resubmission. Equations must be numbered sequentially.	
Please check whether your manuscript contains third-party images, such as figures from the literature, stock photos, clip art or commercial satellite and map data. We strongly discourage the use or adaptation of previously	

published images, but if this is unavoidable, please request the necessary rights documentation to re-use such material from the relevant copyright holders and return this to us when you submit your revised manuscript. An appropriate permissions statement must be present in the relative figure caption for any third-party images.	
Please check that you have not copied any text directly from published work (even your own) without clear attribution, including one or more references. We run a plagiarism detection software and may need to request additional changes if we identify large blocks of identical text.	
An updated editorial policy checklist that verifies compliance with all required editorial policies must be completed and uploaded with the revised manuscript. All points on the policy checklist must be addressed; if needed, please revise your manuscript in response to these points. https://www.nature.com/documents/nr-editorial-policy-checklist.pdf. Please note that this form is a dynamic 'smart pdf' and must therefore be downloaded and completed in Adobe Reader. This file will not open in an internet browser.	
The reporting summary will be published alongside your manuscript therefore it needs to accurately represent your work. In this case, please take a closer look at the reporting summary and make sure things are completed correctly. If an item does not apply, for example human participants, I need you to check the NA box next to that item. No section should be left blank. Also, please make sure to include your name and date at the top of the document.	

If you require a new Reporting Summary form, please download it here: https://www.nature.com/documents/nr-reporting-summary.pdf. Please note that this form is a dynamic 'smart pdf' and must therefore be downloaded and completed in Adobe Reader. This file will not open in an internet browser.	
Your paper will be accompanied by a brief editor's summary when it is published on our homepage. Please approve the draft summary below or provide us with a suitably edited version (no more than 250 characters including spaces). Two engineering approaches, a rotating cell culture system (RCCS) and microfluidic platform (μ platform), are presented with the goal of improving harvestability, scalability, reproducibility and survival of human brain organoids in long term culture.	
ORCID Communications Biology is committed to improving transparency in authorship. As part of our efforts in this direction, we are now requesting that all authors identified as 'corresponding author' create and link their Open Researcher and Contributor Identifier (ORCID) with their account on the Manuscript Tracking System (MTS) prior to acceptance. ORCID helps the scientific community achieve unambiguous attribution of all scholarly contributions. For more information please visit http://www.springernature.com/orcid. For all corresponding authors listed on the manuscript, please follow the instructions in the link below to link your ORCID to your account on our MTS	

before submitting the final version of the manuscript. If you do not yet have an ORCID you will be able to create one in minutes. https://www.springernature.com/gp/researchers/orcid/orcid-for-nature-research IMPORTANT: All authors identified as ‘corresponding author’ on the manuscript must follow these instructions. Non-corresponding authors do not have to link their ORCIDs but are encouraged to do so. Please note that it will not be possible to add/modify ORCIDs at proof. Thus, if they wish to have their ORCID added to the paper they must also follow the above procedure prior to acceptance. To support ORCID's aims, we only allow a single ORCID identifier to be attached to one account. If you have any issues attaching an ORCID identifier to your MTS account, please contact the Platform Support Helpdesk at http://platformsupport.nature.com/	
We regularly highlight papers published in Communications Biology on the journal’s Twitter account (@CommsBio). If you would like us to mention authors, institutions, or lab groups in these tweets, please provide the relevant twitter handles in the right-hand column.	
We would welcome the submission of material for the ‘Featured Image’ section on the Communications Biology home page. Images should relate to the content of your manuscript but need not be contained within the paper. Photographs and aesthetically interesting images are preferred; diagrams are generally not used. Suggestions should be uploaded as a Related Manuscript file. Please provide 1200x675-pixel RGB images. You will also need to submit a completed Image License to Publish. Unfortunately, we cannot promise that your suggestions will be used.	

Supplementary information	
Supplementary Information Format and referencing  ● Supplementary Figures, small Tables, and any supplementary text must be provided in a single PDF. Figures and their captions should be presented together.  ○ If you include a title page, please check that the title and author list matches the main manuscript. ● All Supplementary items must be referred to in the manuscript, and items must be mentioned in numerical order. Please do not include general references to “Supplementary Material”; instead refer to specific items. ● Additional files can be provided as Supplementary Data (Excel files, text files, .zip folders), Supplementary Movies, Supplementary Audio, or Supplementary Software (.zip folder) Supplementary Information files will be uploaded with the published article as they are submitted with the final version of your manuscript. Any highlighting or tracked changes should be removed from the file.	All supplementary files have been overhauled, the title and author list has been added, excel files that were uploaded previously, as "Supp. Table 2-3" have been revised and labelled as "Supp. Data 2-3". Other table numbers are also revised in the manuscript and related files. Finally, main supplementary file is provided in a single PDF.
Supplementary items must be cited in a consistent format. Names of items in the Supplementary file(s) must match those used in the main manuscript. We recommend using the following naming formats: Supplementary Figure 1, Supplementary Table 1, Supplementary Data 1, Supplementary Note 1, and Supplementary References.	

Large tables and other data types: We strongly recommend depositing these to suitable repositories (such as Figshare, Dryad, or a data type-specific repository if one exists). Otherwise, these must be supplied as Supplementary Data files. Each file must be labelled as Supplementary Data 1, etc.	The excel files that were uploaded previously, as "Supp. Table 2-3" have been revised and labelled as "Supp. Data 2-3". Finally, each file is provided as an excel file.
It's mandatory to provide access to the numerical source data for graphs and charts: We strongly recommend depositing these to suitable repositories (such as Figshare, Dryad, or a data type-specific repository if one exists). Otherwise, all source data underlying the graphs and charts presented in the main figures must be uploaded as Supplementary Data (in Excel or text format). Note that only the data used directly for generating the charts needs to be supplied.	The numerical source data for Fig. 2b,c,d,e,f,g, Fig. 5d and Fig. 6d graphs are added in excel file and provided as Supplementary Data 1. Thus, all graphs are revised in Figures and all figures are revised in the manuscript.
For any Supplementary Files such as those mentioned above that are not included your combined PDF (e.g. Supplementary Data, Movies, Audio, Software), please provide a title and description for each file here in the column to the right. For example: File name: Supplementary Data 1 Description: The source data behind the graphs in the paper	File name: Supplementary Data 1 Description: The numerical source data for a percentage of organoid harvestability results (Fig. 2b graph), b plots of organoid sizes (Fig. 2c,d,e,f,g graphs), c extracellular glutamate concentrations of organoids (Fig. 5d graph), d percentage of apoptotic zone (Fig. 6d graph) File name: Supplementary Data 2 Description: Biological process (Gene Ontology) enrichment dataset of upregulated genes in all maturation systems and time intervals. a Upregulated genes, b Network statistics and c Enriched Biological process

	(GO) terms (uploaded separately as an excel file) File name: Supplementary Data 3 Description: Tissue expression (TISSUE) enrichment dataset of upregulated genes in all maturation systems and time intervals. a Upregulated genes, b Network statistics, and c Enriched Tissue expression (TISSUE) terms (uploaded separately as an excel file)
Title Page	
Please ensure that the author list provided in our manuscript tracking system matches the author list in the main manuscript.	
Please check that your author list and affiliations comply with the following:  ● Where relevant, “present address” must be provided separately as the final affiliation. ● At least one corresponding author must be designated, and an e-mail address must be provided for each corresponding author (with a limit of one e-mail address per author). 	
Manuscript title Please ensure the title clearly describes the central finding of the paper. We recommend writing the title as a declarative statement of approximately 15 words or fewer.	

Be sure to include any key species, protein names, or gene names to ensure optimal retrieval of the paper in database searches.	
Abstract The abstract should be accessible to non-specialists and avoid jargon and abbreviations. Please write the abstract in the present tense. We recommend structuring the abstract as follows: Opening statement explaining why this area of research is important. A sentence explaining the gap in knowledge that your research will address. Here we show (or an equivalent phrase), and then the major results and conclusions of the paper. Final sentence indicating any broader impacts and how this research will be used in the future.	
Main text	
Format of the main text Please ensure your manuscript includes the following sections, presented in this order:  1. “Introduction”: The background and rationale for the work. The final paragraph should be a brief summary of the major results and conclusions. The results of the current study must only be discussed in this final paragraph. The Introduction should contain no references to figures or tables. Do not include subheadings. 2. “Results” or “Results and Discussion”. This should be split into subheaded sections; we recommend 1 subheading per main figure or 	Methods section in manuscript is improved as suggested by including only 1 subheading.

table. Figures should not be embedded in the text but submitted separately.  a. Do not use more than 1 layer of subheadings. b. A “Conclusions” paragraph can be included only if the results and discussion are combined into a single section. 3. “Discussion” (optional), without subheadings. 4. Methods, which should be split into subheaded sections. Do not use more than 1 layer of subheadings. To improve readability, we recommend that the main text (Introduction, Results and Discussion) be limited to approximately 5000 words or fewer.	
Statistical reporting Wherever statistics have been derived (e.g. error bars, box plots, statistical significance) the legend needs to provide and define the n number (i.e. the sample size used to derive statistics) as a precise value (not a range), using the wording “n=X biologically independent samples/animals/independent experiments” etc. as applicable.	Thank you very much for the suggestions, the necessary arrangements have been made in the manuscript.
Statistical representation Statistics such as error bars cannot be derived from $n < 3$ and must be removed from all such cases. We strongly discourage deriving statistics from technical replicates, and they should be removed from all such cases, unless there is a clear scientific justification for why providing this information is important. Conflating technical and biological variability, e.g. by pooling technically replicate samples across independent experiments is strongly discouraged.	In order to avoid confusion, the plots are revised, the necessary arrangements are made in the manuscript and highlighted.

Please include exact p-values where possible. We ask that you also include the name of the statistical test and the estimated effect size. If applicable, please also include the confidence interval.	
Avoid the use of the word “significant” unless referring the results of a statistical test.	
Please check that all gene and mRNA names are in italics. Protein names should not be in italics. Please confirm that only official gene/protein symbols are used and that species names are in italics.	
All data that support the conclusions drawn must be presented in the manuscript unless they are published elsewhere. We do not allow statements of “data not shown”.	We removed this part in the manuscript as suggested
Please avoid abbreviating terms unless they are used five or more times. We ask that you avoid all non-standard 2 letter abbreviations.	
Use of speech marks around words or phrases should be avoided; if a phrase is non-standard, please explain the meaning instead; otherwise they are usually unnecessary.	
Display items	
Figure captions/legends Figures must have a title that will appear above the Figure and a legend that will appear below the Figure (see e.g. https://www.nature.com/articles/s42003-020-1059-1/figures/1)	

The Figure title must describe the Figure as a whole and must not contain reference to specific figure panels. The Figure legend must refer to and describe all panels. Abbreviations, symbols, colors, and shading present in the Figure must be defined. Please write out the symbols/colors in words (blue circles, red dashed line, etc.) within these definitions. All figure panels must be labelled using lower case letters. Please refrain from referring to sections of figures as top/bottom/left/right/, etc.	
Axis and panel labels will be published as received. We recommend using a sans-serif font such as Arial or Helvetica.	
Data presentation in bar graphs and line graphs For all graphs depicting a single point value (e.g., mean) with error bars, you must add individual data points or convert the graph to a boxplot or dot-plot. You may wish to refer to this blog post about representing data distribution in plots (particularly for small datasets). We strongly encourage the same for plots with multiple time courses depicted. See the June 24, 2019 CommsBio editorial for more details about this policy. Example plots are shown here:	We converted Fig. 2b, Fig. 5d and Fig. 6d graphs to dot-plot like graphs as suggested.

Examples of plots showing data distribution. Figure 2 from the editorial linked to above.

When choosing a color scheme please consider how it will display in black and white (if printed), and to users with color blindness. Please consider distinguishing data series using line patterns rather than colors, or using optimized color palettes such as those found at <https://www.nature.com/articles/nmeth.1618>.

The use of colored axes and labels should be avoided.

Please avoid the use of red/green color contrasts, as these may be difficult to interpret for colorblind readers.

Please define the error bars in each Figure and Supplementary Figure where they are used. One statement at the end of each Figure caption is sufficient if the error bars are equivalent throughout the Figure.	
Microscopy images and photographs in each Figure and Supplementary Figure must be accompanied by scale bars, and these must be defined.	
Blots and gels All blots/gels must be accompanied by size markers in every figure panel. Uncropped and unedited blot/gel images must be included as Supplementary Figure(s). The new Supplementary Figure(s) must be cited in the main manuscript text (for example, in the Data Availability Statement). Please pay close attention to our Digital Image Integrity Guidelines and to the following points below:  ● that unprocessed scans are clearly labelled and match the gels and western blots presented in figures. Unprocessed scans must be included in a supplementary figure. ● that control panels for gels and western blots are appropriately described as loading on sample processing controls ● all images in the paper are checked for duplication of panels and for splicing of gel lanes. Finally, please ensure that you retain unprocessed data and metadata files after publication, ideally archiving data in perpetuity, as these may be requested during the peer review and production process or after publication if any issues arise.	Within the scope of this study, WB analysis was performed for depiction of 22 different proteins (CD31, CD11b, NEUN, GFAP, MBP, Nestin, Ncad, CTIP2, SATB2, TBR2, TBR1, FOXG1, TUJ1, PAX6, SOX2, PSD95, PROX1, GABA-A, vGLUT1, Ki-67, b-catenin and b-actin) for a total of 8 different sample groups (shaker spinner, RCCS, and μ-platform organoids on d60 and d120). Within the comments, the original blot images of the WB analysis of these proteins-groups are given as Supp. Fig. 2, and the manuscript is revised according to the changes. The following should be considered when viewing blot images: *The protein standard (Bio-Rad, 161-0394) used in WB analysis while gel loading, could not be effective for visualization together with protein bands in chemiluminescence

communications biology

imaging system (Bio-Rad, ChemiDoc MP Imaging System containing a CCD camera) at a wavelength of 428 nm. Therefore, the kDa data of the protein bands were indicated by arrows.

*As a result, although precisely the same amounts of protein were loaded onto the gels for each sample groups, the proteins that freshly isolated from different organoids harvested in different batches at different times, were sometimes loaded onto the **different gels at different times and at different orders.**

*And some results are from experimental groups run on gels loaded with proteins of static organoids and/or day 30 organoids whose results are not shown in the study. For that, the orders of samples given in some membranes are also different.

*On the other hand, most of the time, **PVDF membranes were cut at places close to the recommended kDa sizes** for targeted proteins, and effective labeling were made with a small amount of antibodies in order to study repetitively and economically.

	*Also sometimes, proteins with around the same kDa are labeled on the same cutted membrane after effective blocking.
Tables in the main text Please check that your Tables comply with the following:  • Do not include shading or colors. All Tables must contain black and white text only. • Any bold/italic formatting must be either removed or defined clearly in a Table footnote. • Where Tables contain images, each image should appear in its own cell in the absence of any text. • All Tables must have a brief title. 	
Methods	
Please ensure that all information present in the Reporting Summary is also in the manuscript. This information is usually most appropriate in the Methods section.	
We allow unlimited space for Methods. The Methods must contain sufficient detail such that the work could be repeated. It is preferable that all key methods be included in the main manuscript, rather than in the Supplementary Information. Please avoid use of “as described previously” or similar, and instead detail the specific methods used with appropriate attribution.	

The Methods should include a separate section titled “Statistics and Reproducibility” with general information on how the statistical analyses of the data were conducted, and general information on the reproducibility of experiments (also those lacking statistical analysis), including the sample sizes and number of replicates and how replicates were defined.	We revised “Statistical analysis” section as “Statistics and Reproducibility” and detailed in manuscript as recommended.
We encourage you to include the following statement about the use of human brain organoids in your studies: All experiments involving hiPSCs from human subjects were performed in compliance with [please provide ethical review committee and any reference numbers]. Please also consider including one of the following statements: [ ] The authors concluded [in consultation with, or not in consultation with] ethicists that no testing for sentience or sentience-like capacities was necessary for this experiment; or [ ] The authors implemented the following measures to test or address any potential questions about sentience or sentience-like capacities in this experiment	For this study, we used iPSC line that was previously derived from fibroblasts of a healthy donor and characterized by our project partners during their previous study that was published in Stem Cell Reports Journal (https://doi.org/10.1016/j.stemcr.2019.03.007). Therefore, there were no ethical review committee requirements for this study, which was revised as follows; “Maintenance of human induced pluripotent stem cells (iPSCs). iPSC lines that were previously reprogrammed from human dermal fibroblasts of healthy donors and characterized in terms of pluripotency markers and mycoplasma purity, were obtained from Izmir Biomedicine and Genome Center, Stem Cell and Organoid Technologies Laboratory¹¹⁵. So, there were no ethical review committee requirements for this study.”

If applicable, all oligo sequences, concentrations of antibodies, and sources of cell lines must be included in the Methods (these can also be provided in a main Table and cited in the Methods).	
Nature Portfolio journals encourage authors to share their step-by-step experimental protocols on a protocol sharing platform of their choice. The Nature Portfolio’s Protocol Exchange is a free-to-use and open resource for protocols; protocols deposited in Protocol Exchange are citable and can be linked from the published article. More details can be found at https://protocolexchange.researchsquare.com/	
Data Policies	
The Data Availability statement must include:  • Access details for deposited data, including repository name and unique data ID. • How source data can be obtained. • A statement that all other data are available from the corresponding author (or other sources, as applicable) on reasonable request. Note that ‘available upon request’ is only appropriate if immediate data access has not been mandated by our policies or by the editors. See here for more information about formatting your Data Availability Statement: http://www.springernature.com/gp/authors/research-data-policy/data-availability-statements/12330880	Data availability section was revised as follow: Source data for the presented figures are provided as Supp. Data 1-2-3 with this paper. Further simulation data that is generated and analyzed during the current study is available from the corresponding author on reasonable request. Besides that, the normalized RNA expression levels of the specific neuronal/glial cell markers in different cell types downloaded from RNA single cell type data from Human protein atlas database; https://www.proteinatlas.org/about/download;rna_single_cell_type.tsv.zip. Also, Biological Process (Gene Ontology) and Tissue expression (TISSUES) enrichments of

	upregulated genes of each system and day groups were realized via STRING v11.5 database (https://string-db.org/)
Mandatory deposition of raw and processed data is required for: ● All sequencing data (DNA, RNA, protein)● Novel human genetic polymorphisms (e.g., dbSNP)● Linked genotype and phenotype data (e.g., dbGaP for human data)● GWAS summary statistics or polygenic risk scores● Novel macromolecular structure● Gene expression microarray data (must be MIAME compliant)● Crystallographic data for small molecules● Mass spectrometry-based proteomics data For more information on mandatory data deposition policies at the Nature Portfolio, please visit http://www.nature.com/authors/policies/availability.html#data For an up-to-date list of approved repositories for each mandatory data type, please visit https://www.springernature.com/gp/authors/research-data-policy/repositories/12327124. Accession code(s) for deposited data must be provided in the Data Availability statement in the final version of the paper. Failure to do so will delay publication. Please ensure data are available prior to publication.	

Communications Biology has a strong preference for all data to be deposited in an approved repository. In some cases, data deposition may be required by the editor.

We recommend the following data repositories:

- GenBank (all DNA sequence data)
- NHGRI-EBI GWAS Catalog (GWAS summary statistics)
- PGS Catalog (polygenic risk scores)
- Gene Expression Omnibus (Microarray or RNA sequencing data)
- Sequence Read Archive (WGS or WES data)
- Protein Data Bank (protein structural data)
- OSF (neuroimaging raw data and EEG/EMG/MEG raw data)
- Neurovault (unthresholded statistical maps, parcellations, and atlases produced by MRI and PET studies)
- Image Data Resource (microscopy data)
- PRIDE (proteomics data)

Data types without a specific repository can be deposited in a generalist repository, such as figshare or Dryad.

For an up-to-date list of approved repositories, please visit <https://www.springernature.com/gp/authors/research-data-policy/repositories/12327124>.

Data citation

Please cite datasets stored in external repositories **in the main reference list**.

For previously published datasets, we ask authors to cite both the related research articles and the datasets themselves.

For more information on how to cite datasets in submitted manuscripts, please see our data availability statements and data citations policy.	
Please deposit your newly generated plasmids in a community repository (eg, Addgene). Include the ID numbers in the Data Availability statement.	
Code availability Please include a Code Availability statement, indicating whether and how the code can be accessed, including any restrictions to access. In some cases, the editor may require that code be made immediately available. This section should also include information on the versions of any software used, if relevant, and any specific variables or parameters used to generate, test, or process the current dataset. The Code Availability statement must be provided as a separate section after the Data Availability section. Please see our policy on code availability for more information. http://www.nature.com/sdata/for-authors/editorial-and-publishing-policies#code-avail In addition to making the custom code available, please ensure that the version of the code/software described in the paper is deposited in a DOI-minting repository (eg, Zenodo) and that this DOI is also cited in the main Reference list.	We added Code availability section to the manuscript file: “Details of publicly available software used in the study are given in the “Methods and Data availability” section. Apart from this, no special custom code or mathematical algorithms were central to reaching the conclusions of this work.”
End Notes	
Please check that your bibliography complies with the following:	

 ● Your bibliography should start with the heading “References”. The references must be numbered in the order of appearance in the text, then tables, then figures. ● Any in-text citations to references (e.g. "Gupta et al. show...") should be followed by their corresponding reference citation number from the reference list. ● Manuscript citations must include journal title, article title, volume number, page or article number or DOI, and year of publication. ● No publication can be present more than once in the reference list. ● No footnotes are permitted in the references or elsewhere. Text should be incorporated into the main text, the Methods section, or the Supplementary Information instead. ● Websites should only be listed in the references if they are in common use or curated. ● Where possible, preprints in the reference list should be updated with details of the published, peer-reviewed paper. ● Citations should be formatted in the text using superscript numbers. 	
Please provide a Competing interests statement using one of the following standard sentences:  ● The authors declare the following competing interests: [specify competing interests] ● The authors declare no competing interests. See our competing interests policy for further information: https://www.nature.com/nature-research/editorial-policies/competing-interests	We added Competing interests section at the end of the manuscript file: “The authors declare no competing interests.”
Please check that your Author Contributions section individually lists the specific contribution of each author to the work. Each author must be	We checked the Author Contributions section carefully and added specific

referred to by name or initials. Where multiple authors possess identical initials, they must be clearly disambiguated from one another. See our author contributions policy for further information: https://www.nature.com/nature-research/editorial-policies/authorship#author-contribution-statements	contribution of each authors (total 11 authors) to the work by initials (as P.S-M., U.D., Y.F., S.A., G.B., B.G., B.Y., S.Y., C.B-A., E.E., O.Y-C.).
No separate funding section is permitted. Please include your funding information in Acknowledgements instead.	We included our financial funding information in the Acknowledgements section.